# Exploring Water Accumulation Dynamics in the Pearl River Estuary from a Lagrangian Perspective

Mingyu Li[1], Alessandro Stocchino[2,3], Zhongya Cai[1,3], Tingting Zu[4]

[1]State Key Laboratory of Internet of Thing for Smart City, Department of Ocean Science and Technology, University of Macau, Macau, 999078, China

[2]Department of Civil and Environmental Engineering, The Hong Kong Polytechnic University, Hong Kong, 999077, China

[3]Center for Ocean Research in Hong Kong and Macau (CORE), Hong Kong, 999077, China

[4]State Key Laboratory of Tropical Oceanography, South China Sea Institute of Oceanology, Chinese Academy of Sciences, Guangzhou, 510301, China

*Correspondence to*: Zhongya Cai (zycai@um.edu.mo); Tingting Zu (zutt@scsio.ac.cn)

**Abstract.** Water accumulation is essential for understanding estuarine mass distribution and ecosystem management. In this study, we examined the water accumulation dynamics in the Pearl River Estuary (PRE) from a Lagrangian perspective. Generally, there is a notable negative correlation between the horizontal velocity divergence ( $\nabla_h \vec{V}_h$ ) and the accumulation probability. Influenced by density fronts and velocity convergence, we observed significant bottom-layer accumulation of particles in the western estuary and Hong Kong waters during summer, whereas the accumulation moved landward in winter. Sub-regions with distinct accumulation patterns and interconnections were identified and combined with the trajectories. In summer, the western estuary and Macau waters have a substantial net negative $\nabla_h \vec{V}_h$ and strong density fronts are major accumulation targets, which attract particles from the whole estuary. Conversely, the eastern estuary and Hong Kong waters exhibit significant westward motion, influencing the western side. In winter, particles are more likely to accumulate in their original locations. The upper estuary becomes a major accumulation area because of the obstructive density front, and decreased river discharges. The tidal currents and river discharges mainly control water accumulation in the estuary by changing the mixing or current intensity. The weakening of tidal currents and river discharges induced the intensified bottom intrusion and the landward movement of accumulation.

## 1 Introduction

The Pearl River Estuary (PRE), located in the northern South China Sea (NSCS) (Fig. 1a), is influenced by the East Asian Monsoon, with northeasterly winds prevailing in winter and southwesterly winds prevailing in summer (T. Li & Li, 2018). Thus, in the PRE, winter is characterized as a dry season, and summer is characterized as a wet season due to the large rainfall induced by the moist air brought from the South China Sea; consequently, the river discharge in summer (~ 20,000 $m^3s^{-1}$) is approximately five times more than that in winter (~ 3,600 $m^3s^{-1}$) (Harrison, Yin, Lee, Gan, & Liu, 2008). This is quite different from many other river deltas, such as the Mississippi deltas, where river discharge reaches a maximum in winter and spring, but is reduced in summer and autumn (Lane et al., 2007).

As a bell-shaped estuary, the width increases from approximately 5 km at the upper end to 35 km at the lower end. Despite the two narrow, deeper channels (~20 m in depth), the PRE is shallow, with a water depth of approximately 2-10 m. The PRE is a partially mixed estuary in which circulation is jointly controlled by river discharge, tides, wind, and topography (Ascione Kenov, Garcia, & Neves, 2012; Banas & Hickey, 2005; Gong, Shen, & Hong, 2009; C. He, Yin, Stocchino, & Wai, 2023; C. He, Yin, Stocchino, Wai, & Li, 2022; Liu, Zu, & Gan, 2020). There are two distinct dynamic regimes in the PRE. The narrow upper part of the PRE shows classical gravitational circulation, whereas in the wider lower part of the PRE, where the Coriolis effect becomes essential, the topography and interaction with the monsoon-driven shelf current complicate the circulation (Dong, Su, Ah Wong, Cao, & Chen, 2004; Wong et al., 2003; Zu & Gan, 2015). Gravitational circulation occurs in the two deep channel regions, whereas currents show precise seasonal characteristics over the shallower western estuary. Geostrophic wind-driven coastal currents intrude into the PRE during the summer upwelling season (Zu & Gan, 2015), whereas seaward buoyancy-driven coastal currents flow out of the PRE during winter(Dong et al., 2004; Wong et al., 2003). The alternation of the spring-neap tide and variation in river discharge play crucial roles in modulating stratification and mixing inside the PRE (Mao, Shi, Yin, Gan, & Qi, 2004; Pan, Lai, & Thomas Devlin, 2020; Zu, Wang, Gan, & Guan, 2014). Strong tidal mixing in the middle PRE has led to the conversion of estuarine river plumes into buoyancy-driven coastal currents (Dong et al., 2004; Zu et al., 2014).

In recent decades, the health and sustainability of estuaries have been increasingly imperiled by environmental issues caused by anthropogenic activities and climate variability (Dai et al., 2006; Fok & Cheung, 2015; Shen, Wei, Shi, Gao, & Zhou, 2023). Estuarine circulation and the associated mass transport are critical for addressing these environmental challenges. The specific areas of water accumulation and the extent of its exchange with the open sea significantly impact the dispersion of oceanic pollutants and biogeochemical health (Acha, Mianzan, Guerrero, Favero, & Bava, 2004; Hinojosa, Rivadeneira, & Thiel, 2011). In estuaries, some regions are more likely to attract water because of complicated current

circulation, which can be considered as stronger horizontal convergence targets for some materials (T. Wang et al., 2022).

For example, the salt wedge acts as a significant pollutant sink in an estuary, and a higher concentration of microplastics is always obtained at the salt wedge in the Rio de la Plata estuary (Acha et al., 2003; Vermeiren, Muñoz, & Ikejima, 2016). Areas accompanied with higher concentration of nitrogen and phosphorus are usually appearing eutrophic (Tao, Niu, Dong, Fu, & Lou, 2021). Heavy metal pollution in estuaries has been observed in areas that prefer to concentrate fine particles (Balachandran et al., 2005). Therefore, identifying the accumulation areas in estuary-shelf systems provides an adequate

estimate for surveying pollutant sinks(Mestres et al., 2006; Tao et al., 2021; Vermeiren et al., 2016; A.-j. Wang et al., 2016). With intensified human activities, pollutant sinks related to the accumulation phenomena in the PRE have attracted attention. Tao et al. (2021) revealed that the upper part of the PRE is a target sink for nitrogen and silicate. D. Zhang et al. (2013) found that trace elements prefer to accumulate on the PRE's west side. Higher concentrations of microplastics have been observed in western estuaries and Hong Kong waters (Lam et al., 2020). Similarly, studies on hypoxia have shown

that the convergence of buoyancy-driven currents and wind-driven shelf flows contributes to the formation of stable water columns, providing favorable conditions for the development of hypoxic zones (e.g., D. Li et al. (2021); X. Li et al. (2020)). The high frequency of hypoxia in the estuary during summer is related to the strong stratification of the water column (Y. Cui, Wu, Ren, & Xu, 2019; H. Zhang & Li, 2010). These accumulation patterns in the PRE are more concerned with the measurement of pollutant concentrations (Tao et al., 2021), estimation of the pollutant accumulation rate (L. Zhang et al.,

2009), and discussion of the sources of pollutants (Ye, Huang, Zhang, Tian, & Zeng, 2012), and lack a discussion on the understanding of accumulation spatial patterns and underlying physical control.

The Lagrangian tracking method provides an effective way to analyze the transport processes in physical oceanography(Jalón-Rojas, Wang, & Fredj, 2019; van Sebille et al., 2018). It analyzes sizeable virtual particle trajectories calculated from simulated Eulerian three-dimensional and time-varying velocity fields, which capture complex real-world

dynamical processes. Their applications extend across various domains, including interocean exchange(Haza, Özgökmen, & Hogan, 2016), pathway analysis (Jalón-Rojas et al., 2019), and the impact of ocean currents on ecosystems (Chenillat et al., 2015; Dawson, Sen Gupta, & England, 2005; Lebreton, Greer, & Borrero, 2012; Paris et al., 2012).

In this study, Lagrangian tracking and analysis were utilized to examine the features of the accumulation regions in the estuary-shelf system of the PRE and to explore the role of different forcing factors. These results help deepen our

understanding of the environmental effects of multi-scale processes. The remainder of this paper is organized as follows. Section 2 presents the Lagrangian model and numerical solutions of the Markov Chain. Findings regarding the accumulation region and transport connections in the PRE are discussed in Sect. 3. Section 4 presents the roles of hydrodynamic factors. Finally, Section 5 presents the conclusions of this study.

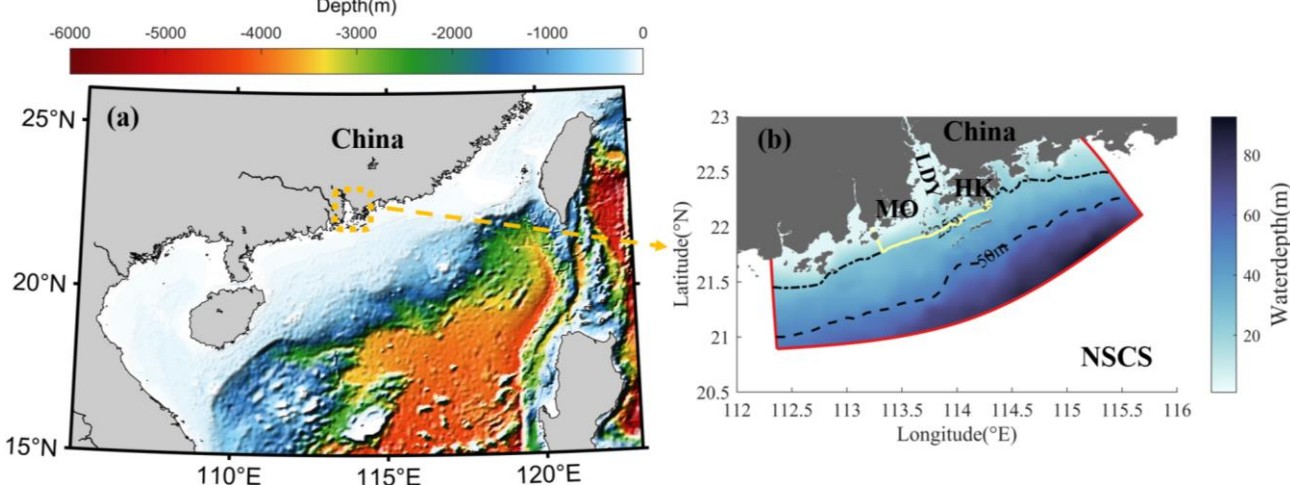

**Figure 1: (a) Location and topography of the Pearl River Estuary (PRE) and adjacent shelf. (b) The bathymetry of the PRE and adjacent shelf. The black dotted and dashed lines represent the 25 m and 50 m isobath. The yellow line defines the seaside boundary of the PRE. LDY, MO, HK and NSCS represent Lingdingyang, Macau, Hong Kong and northern South China Sea, respectively. The red line represents the model boundary.**

## 2 Methods

### 2.1 Lagrangian particle tracking

In this study, multi-scale circulation in the PRE and adjacent shelf regions was simulated by implementing a Regional Ocean Modeling System (Shchepetkin & McWilliams, 2005). The model region covers the estuary and adjacent shelf between 112.3°E-115.68°E and 20.89°N-23.13°N (Fig. 1b). The model used an orthogonal curvilinear grid, and the resolution gradually increased from approximately 1 km over the shelf to ~ 200 m inside the estuary. In the vertical direction, we used the terrain-following S-ordinate (Song & Haidvogel, 1994) to discretize the water column into 30 levels.

The monthly average riverine discharge data were obtained from the Ministry of Water Resources of China. During summer and winter, river discharge was approximately 30000 m$^3$/s and 10000 m$^3$/s (Fig. S1), respectively. Wind forcing, heat flux, and precipitation were obtained from ERA5 atmospheric reanalysis data from the European Center for Medium-Range Weather Forecasts (ECMWF) and were used to force ocean circulation through the implementation of the bulk computation algorithm (Fairall, Bradley, Hare, Grachev, & Edson, 2003). The shelf current was obtained from a coarser model with good validation that can cover the North-South China Sea and provide information on the barotropic and baroclinic velocities, temperature, salinity, and sea level along the boundaries of the PRE (Deng et al., 2022). Vertical turbulence and diffusion coefficient are determined by the Mellor-Yamada 2.5 turbulence-closure module (Mellor & Yamada, 1982), which

provides the turbulent mixing coefficient. Tidal current simulations considering forcing along open boundaries were obtained from Zu, Gan, and Erofeeva (2008), which included the tidal harmonic constants ($M_2$, $S_2$, $K_2$, $N_2$, $K_1$, $O_1$, $P_1$, $Q_1$, and $M_4$). The simulation was validated using long-term remote sensing and local observations and has been used in PRE studies (Cai, Liu, Liu, & Gan, 2022; Chu et al., 2022b; L. Cui, Cai, & Liu, 2023; L. Cui, Liu, Chen, & Cai, 2024).

Particle trajectories were traced by a three-dimensional offline Lagrangian TRANSport model (LTRANS v.2b), which captures complicated dynamical processes in the real world using Eulerian flow fields and turbulent mixing from the hydrodynamic model (Chu et al., 2022b; Liang et al., 2021; Elizabeth WEW North et al., 2011). To reasonably calculate the Lagrangian trajectories in the circulation of estuary-shelf systems, the tracking model considers the advection, turbulence, individual behaviors of particles (e.g., vertical sinking, floating, or swimming velocity), settlement, and

boundary behaviors during particle trajectory simulations. The 4[th] order Runge-Kutta scheme was implemented to handle the advective terms and yield accurate particle trajectories (Dippner, 2009). Considering the random walk of water parcels in the ocean, the model adopts different diffusivity coefficients to control the vertical and horizontal turbulence (E. W. North, Hood, Chao, & Sanford, 2006; Zhong & Li, 2006). This model, primarily based on climatological data, was carefully verified using satellite remote sensing and long-term observations to ensure an accurate representation of the hydrodynamic

properties (Fig. S2). Generally, the model captures the seasonal features of circulation in this region and has been used in previous studies (Cai et al., 2022; Chu et al., 2022a; L. Cui et al., 2024).

 In the surface/bottom tracking case, 8386 particles were uniformly distributed at water surface/bottom across the estuary and adjacent shelf with a 0.01degree interval. Particles were released every two days and tracked for 30 days. The hydrodynamic simulation results were stored every 20 min in January and July to drive the particle tracking during summer

and winter, respectively. During trajectory tracking, the time step of particle tracking was 30 seconds, and their locations were recorded every 20 minutes.

**2.2 Markov Chain**

In this study, to investigate the accumulation features and connectivity among different parts of the domain, we implement the Markov Chains statistical analysis to describe the future state of a random variable, which is based on the particle's

original state positions, to compute a proper transition matrix (K. L. Drouin & Lozier, 2019; Kimberley L. Drouin, Lozier, Beron‐Vera, Miron, & Olascoaga, 2022; Miron, Beron-Vera, Helfmann, & Koltai, 2021; Miron et al., 2019; Miron et al., 2017; van Sebille et al., 2018). To this end, we divided the study area into rectangular grids with dimensions of 0.1 degree, and then the probability of particles moving between different grids within the time interval of $dt$ was calculated as

$$p^t = \frac{n_{ij}^{t_0+dt}}{n_i^{t_0}}, \qquad \text{\textemdash} \qquad (1)$$

where $n_i^{t_0}$ represents the number of particles released in grid $i$ at initial time $t$. $n_{ij}^{t_0+dt}$ represents the number of particles arriving at grid $j$ from grid $i$ after interval time $dt$. Thus, $p^t$ varies for different release grids $i$, arrival grids $j$ and periods $(t_0, t_0 + dt)$. Then, the evolution of the initial distribution $(D^{t_0})$ into a future state $(D^{t_0+dt})$ is achieved by vector-matrix multiplication:

$$D^{t_0+dt} = D^{t_0} \times p^t, \qquad \text{\textemdash} \qquad (2)$$

In this study, the $D^{t_0}$ is defined as a uniform initial distribution in the whole domain that $D^{t_0} = [1, 1, 1, \cdots 1]$. The $D^{t_0+dt}$ represents the evolution of the initial condition under complicated hydrodynamic motion. The high values of D indicate that the regions attracted more particles for accumulation. The transport matrix by Markov Chains helps to predict longer timescale transport pathways through limited short-time trajectories and is widely used to explore transport pathways. Connectivity of water parcels between different regions, such as van Sebille et al. (2018) investigated the pathways of significant currents in the Agulhas Current, Kimberley L. Drouin et al. (2022) found that Caribbean and Gulf of Mexico did not involve the water transition between North Brazil Current and 26°N effectively. According to Jönsson and Watson (2016) and Kimberley L. Drouin et al. (2022), the choice of $dt$ does not significantly influence the results of transport patterns. For the present analysis, we selected a time interval ($dt$) of 2 days.

## 3 Results

### 3.1 Accumulation pattern and regional connectivity

We set up a standard real-world case to explore the accumulation pattern and regional connectivity, which considers river discharge, tidal currents, and wind forcing. Before examining the detailed transport structure, we examined the offshore transport speed of the particles after their release. Using all the particle trajectories, Figure 2 shows the percentage of particles remaining inside the PRE during summer and winter. Here, the seaside boundary of the PRE is defined by a 25 m isobath (yellow line in Fig. 1). If particles move beyond the seaside boundary of the estuary, they may return to the study area due to tidal currents and wind forcing. However, once the particles reached the model domain, they will not be backed again. Overall, the decay speed was faster in summer than in winter, with offshore motions at the surface layer always being quicker than at the bottom layer. Using a value of 20% as a threshold (green dotted line in Fig. 2), during summer, approximately 80% of the surface particles exit the estuary in approximately 10 days, whereas the bottom particles take approximately 15 days. In contrast, during the winter season, with less river discharge, the exit time increased to 20 days

(for the surface particles) and 25 days (for the bottom particles). Thus, for the entire domain, after being tracked for 30 days, most of the particles left the seaside boundary, and their trajectories were used in the following analysis. Using the trajectories of the released particles within 30 days, we explored the final evolved state, which was used to quantify the probability that the particles will accumulate in different regions as a result of the complex hydrodynamics of estuarine circulation.

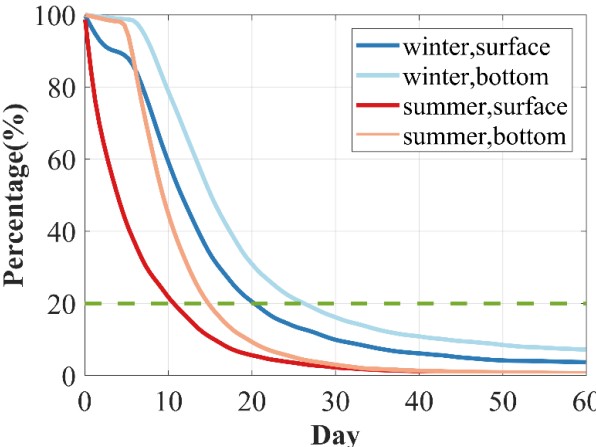

**Figure 2: The changes of particle percentage remaining in the estuary during tracking.**

According to the circulation in the standard case (Fig.3), intensified offshore motions were consistently observed at the surface layer in both summer and winter. Owing to the decrease in river discharge, the offshore current in the upper estuary was weaker during winter than during summer. The shelf current intrudes into the estuary from the west side of Hong Kong and arrives at the middle estuary at the bottom, and the bottom intrusion is more substantial in winter. We selected the AB transect along the intrusion channel to further check the density stratification and vertical circulation (Fig. 4). A large amount of river discharge in summer pushes fresh water to the seaside, leading to seaward movement of seawater (indicated by the isopycnal line of 1015 kg·m$^{-3}$). In winter, the decrease in river discharge caused weaker resistance abilities of fresh water to hinder the intrusion of saltwater.

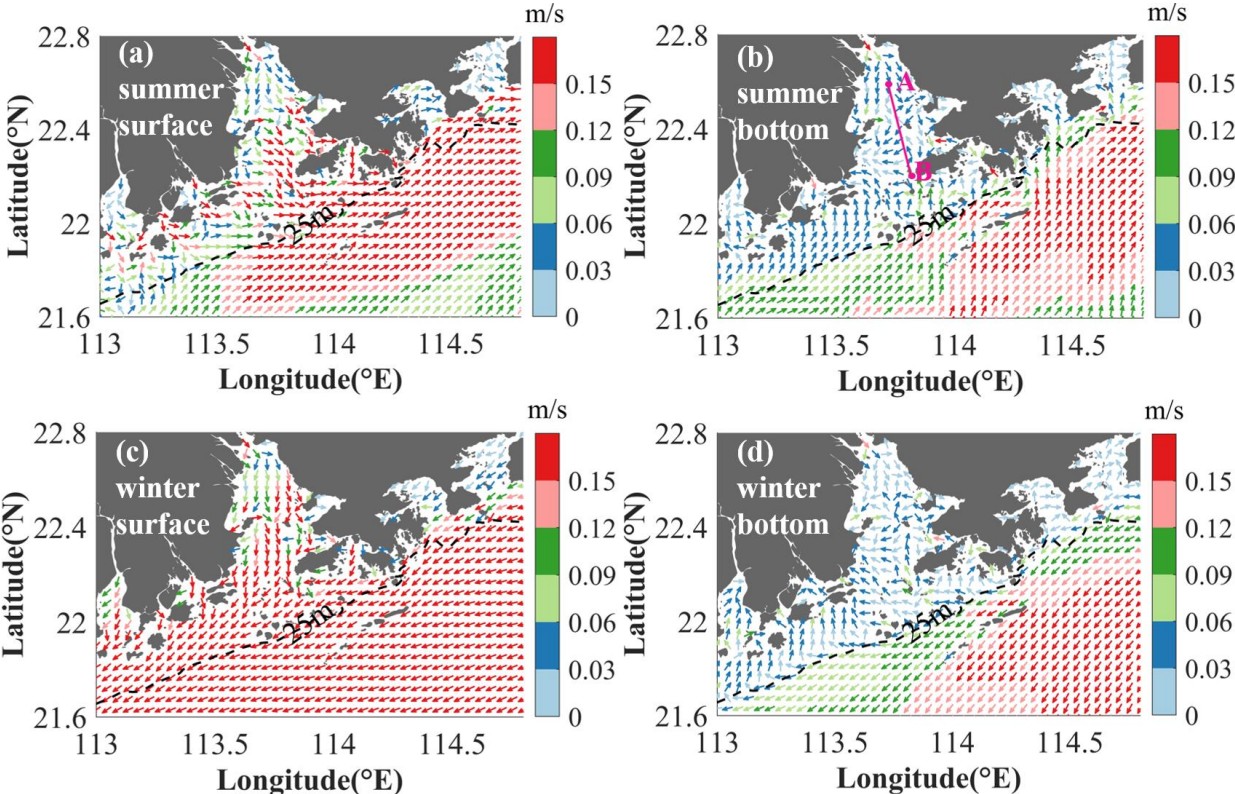

**Figure 3: (a–b) The flow field of the standard case at the surface layer and bottom layer during summertime, respectively. (c-d)**
**are the same as (a-b), but for winter time.**

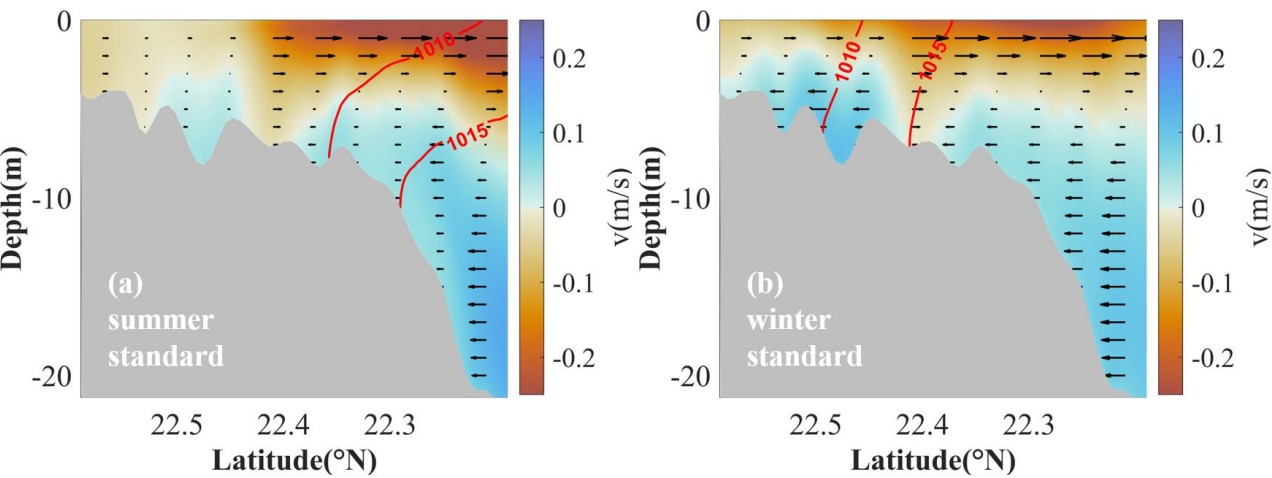

**Figure 4: The along transect velocity (color and arrows, a positive value indicates the onshore intrusion) and density contour of**
**1010 kg·m⁻³ and 1015 kg·m⁻³ (red lines) in AB during summer (a) and winter (b) time. The location of AB is shown in Fig. 3b.**

As a result, compared with the bottom layer, because of the strong offshore current in the surface layer, the surface particles

tended to escape from the estuary quickly, resulting in a very low accumulation probability (Fig. 5a, c). On the contrary, at

the bottom layer (Fig. 5b, d), the spatial distribution of water accumulation was much more apparent. In summer, high

accumulations are observed on the western and southwestern sides (near Macau) and in Hong Kong waters, which are associated with the westward transport current of Hong Kong water and the intensified intrusion current in the lower estuary. Although these regions are close to the open shelf, they are more likely the primary accumulation zones that capture particles initially released in other areas of the domain. Cruise observation data (2014-2018) also showed that these regions generally have a high frequency of hypoxia under the accumulation effect of hydrodynamic processes (D. Li et al., 2021). During winter, with fewer river discharges, gravitational circulation is generally preserved but with weakened intensity. The accumulation regions moved shoreward, particularly in the upper estuary, which is associated with the decreased circulation and landward movement of the dense intrusion. Owing to the intensified northeasterly wind, the westward transport of Hong Kong water and the adjacent shelf current were strengthened, leading to higher accumulations along the west side of the estuary.

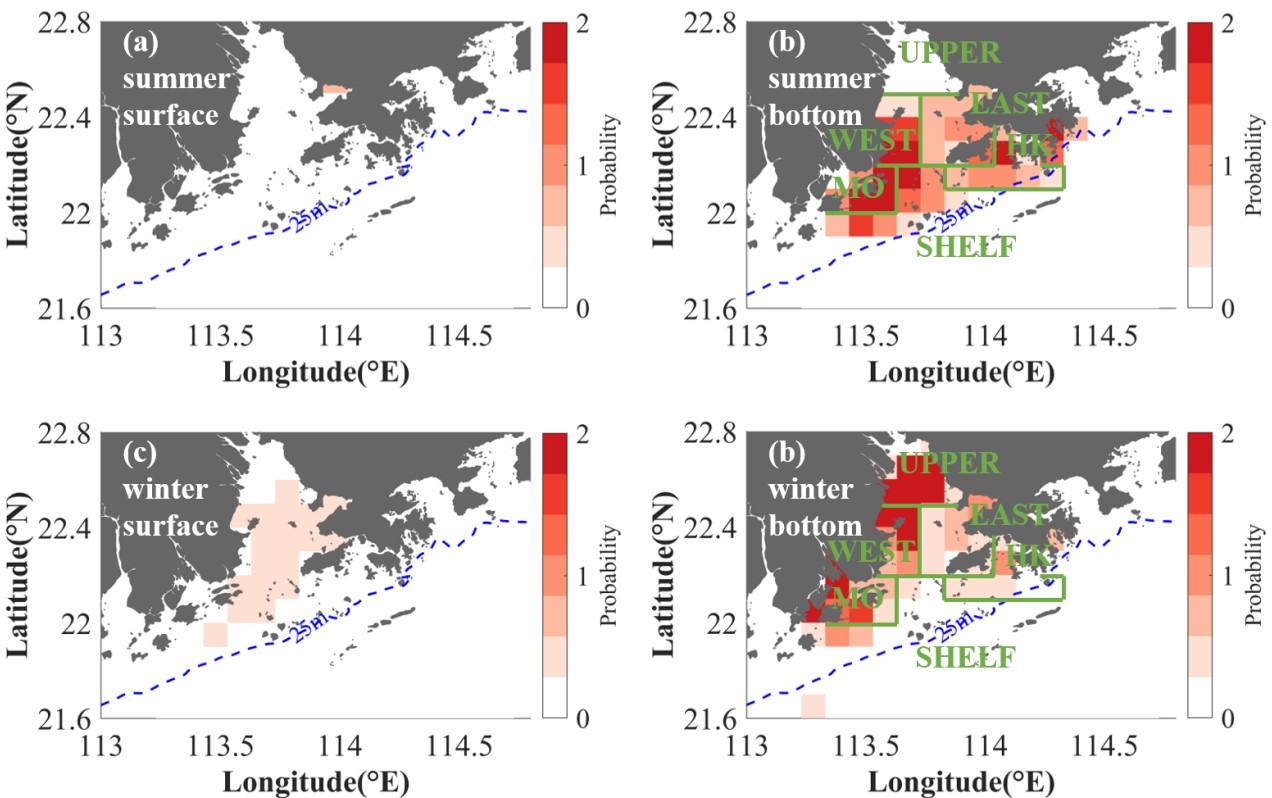

**Figure 5: (a–b) Accumulation probability (color, $D^{t_0}$ in Eq. (2)) at the surface layer and bottom layer during summer time, respectively. The color bar indicates the magnitude of the accumulation probability. (c-d) is the same as (a-b) but winter time.**

Based on the bottom accumulation probability distribution after 20 d, we divide the study area into six subregions: the Upper Estuary (UPPER), Western Estuary (WEST), Eastern Estuary (EAST), Hong Kong Water (HK), Macau Water (MO),

and Shelf Water (SHELF). Compared to the quick offshore motions at the surface layer, the results at the bottom layer were better at illustrating the connectivity between each region. Subsequently, using the trajectories of the particles, we checked the probability of the particles moving in each region during the tracking period (Fig. 6). Sensitivity experiments showed that different tracking periods did not significantly change the significant patterns of the transport matrix.

During summer (Fig. 6a), the original region always made a significant contribution to the accumulation. For the WEST and MO regions, which had the most significant accumulation probability, the accumulated water came from almost all subregions, including the shelf. The onshore intrusion from the shelf water and offshore motions from the upper estuary due to river discharge converge in the water there, as well as westward transport from EAST and HK, facilitating particle accumulation in WEST and MO. In contrast, the EAST and HK regions accumulated particles mainly from the eastern side

of the estuary. Despite the Eulerian currents, there is an onshore intrusion from the shelf and offshore transport from the upper estuary toward these areas.

During winter, the accumulation region moved shoreward and mainly occurred in the UPPER, WEST, and MO regions (Fig. 6b). The particles accumulated in those regions were mainly from the original release regions; thus, it was more difficult for the water to leave the original regions. HK water can affect almost the entire estuary, particularly contributing

to accumulation in the WEST region.

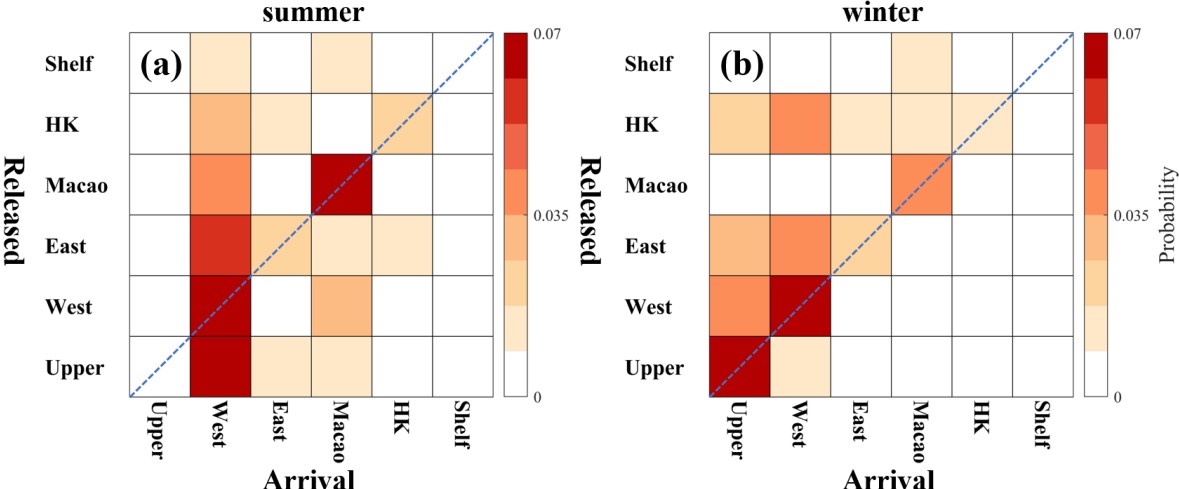

**Figure 6: The connection between six subregions at the bottom layer during the summer (a) and winter (b). The horizontal and vertical axes represented the arrival and release regions, respectively.**

### 3.2 Hydrodynamic Control on the Accumulation

The bottom divergence of the horizontal current, i.e., $\nabla_h \vec{V}_h = \frac{\partial u}{\partial x} + \frac{\partial v}{\partial y}$, which $u$ and $v$ represent the bottom zonal and

meridional velocity, is calculated to examine its influence on the identified bottom accumulation regions (Fig. 7a, b). We established a connection between the average accumulation probability in each subregion and the divergence of the horizontal current. Across the various identified subregions, a substantial negative correlation between the $\nabla_h \vec{V}_h$ and the accumulation probability is observed. There was a correlation coefficient of 0.74 in summer and 0.76 in winter. It suggests that the net negative $\nabla_h \vec{V}_h$, i.e., the convergence of the water provides a favorable condition for the accumulation of water and particles. Such patterns were noted in major accumulation regions, such as the WEST, MO, and UPPER regions, during both summer and winter.

It is also noted that the EAST and HK regions, which show the accumulation of particles, exhibit a weak net positive value of $\nabla_h \vec{V}_h$. This implies that the $\nabla_h \vec{V}_h$ facilitates accumulation under certain conditions, but the actual net accumulation should consider the cumulative effects of velocity convergence along the trajectories, which stay at different locations during the movement. Spatial distribution of $\nabla_h \vec{V}_h$ also illustrates that the intensified negative values occur in the region of the high accumulation probability (Fig. 7c-d), such as the substantial negative $\nabla_h \vec{V}_h$ is observed in MO and WEST regions during summer, as well as in the UPPER region and central part of the HK region during winter. However, the Eulerian perspective of $\nabla_h \vec{V}_h$ presents a complex distribution of alternative positive and negative values. This does not lead to a straightforward identification of the net accumulation areas. Instead, Lagrangian tracking offers a clearer understanding of these regions because it captures the cumulative effects of water motion over time.

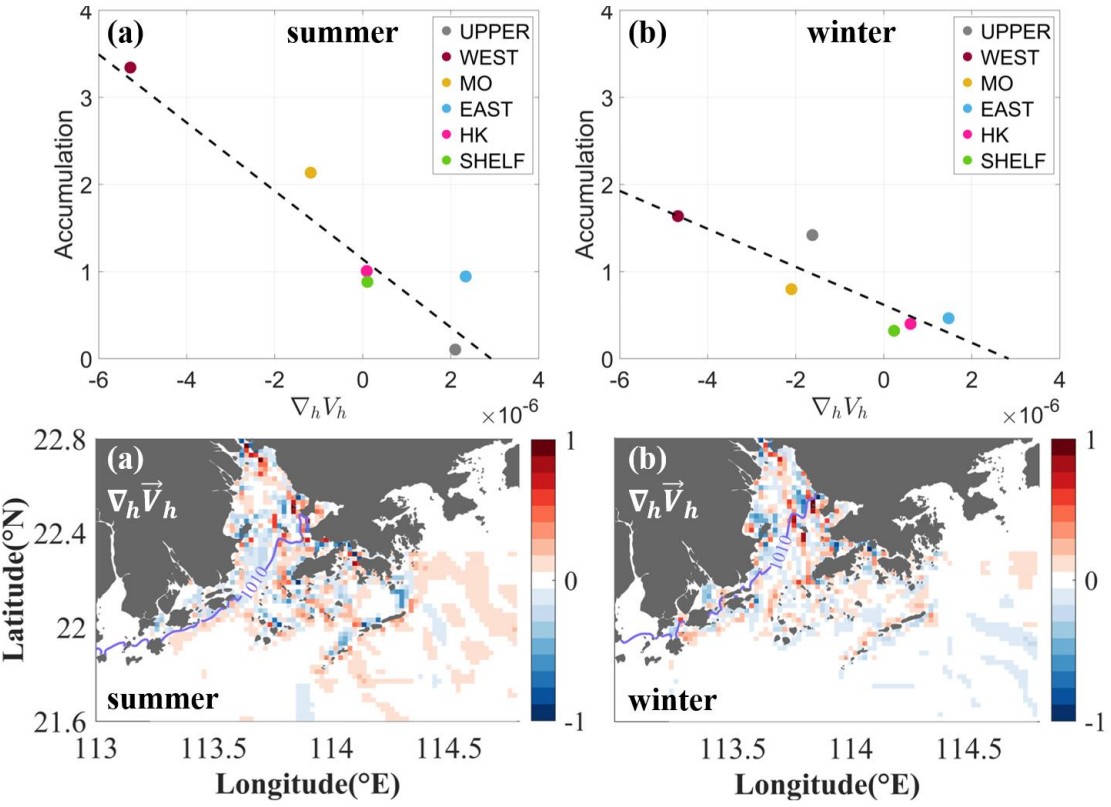

**Figure 7: (a-b)** Scatter plot of accumulation probability against $\nabla_h \vec{V}_h$ for various subregions during summer and winter, respectively. **(c-d)** Horizontal distribution of $\nabla_h \vec{V}_h$ during summer and winter. The $\nabla_h \vec{V}_h$ is normalized by the largest divergence value in this area. Purple lines represented the isopycnic line of 1010 kg·m$^{-3}$.

As a salt-wedge estuary, the existence of a salinity front obstructs particle transport and plays an important role in accumulation regions (Defontaine et al., 2020; Q. He et al., 2018; Vermeiren et al., 2016). The coupling effect of velocity convergence and front makes PRE accumulate particles in the middle estuary and lower estuary during winter and summer, respectively (Malli, Corella-Puertas, Hajjar, & Boulay, 2022). In summer, the heightened river discharge creates the density front ($G = \sqrt{(\frac{\partial \rho}{\partial x})^2 + (\frac{\partial \rho}{\partial y})^2}$) (Fig. 8), which affects the particles transport and promotes their accumulation. The location of

the front, roughly aligns with the outer boundary region of profound negative $\nabla_h \vec{V}_h$ over the WEST and MO region (Fig. 7c), which further supports the accumulation in these regions as the front hinders their offshore movement. During winter, the intrusion reaches the middle estuary, causing a blockage effect on the mass, which tends to remain confined to the upper part of the estuary (Lima, Barletta, & Costa, 2015; Lima, Costa, & Barletta, 2014). This retention likely contributes to the observed accumulation patterns, where the particles remain confined upstream owing to the blockage effect of the front.

The $\nabla_h \vec{V}_h$ in winter also reflects this dynamic that the region of negative values move shoreward.

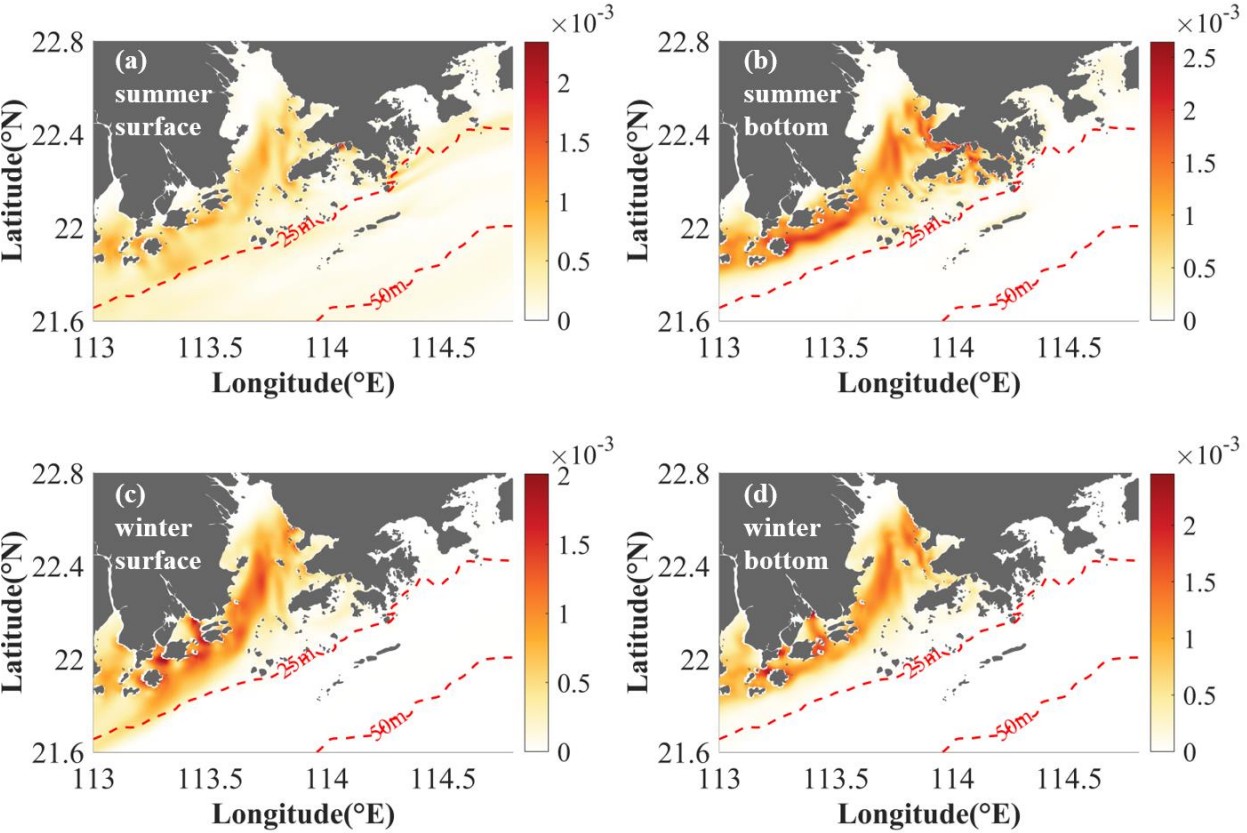

**Figure 8: (a-b) The density front (kg·m$^{-4}$) of the standard case at the surface layer and bottom layer in summer, respectively. (c-d) are the same as (a-b), but for winter.**

## 4 Discussion

The tides and rivers had essential influences on the estuarine circulation and associated mass transport in the PRE, whereas the wind mainly affected the shelf current and had less influence on the mass accumulation inside the estuary (figures not shown here). Two additional experiments were conducted to examine their contributions to the accumulation. In the case

of reduced river discharge, the magnitude of river discharge in the forcing file was reduced by 20%. The magnitude of the tidal current was set to zero to remove the tidal currents. The accumulation regions and connections among the different regions were investigated in the same way using the same Lagrangian tracking and analysis approach.

### 4.1 Tide

Because the tidal current affects the intensity of mixing in the water column, once removed, the offshore motion in the

275 UPPER region is reduced in both summer and winter. The westward transport of Hong Kong water and adjacent shelf currents, which flow south of the WEST region, is strengthened in summer (Fig. 9a). During winter, the intrusion water from the shelf moves straight landward and arrives at the upper part of the estuary. Along the transect of AB (Fig. 3b), we further checked the changes of the density stratification and vertical structure of circulation (Fig. 9c, d), The intensification and landward movement of the bottom intrusion is associated with the stronger density stratification. The removal of the

280 tidal current leads to weaker mixing and intensified stratification, as also revealed in previous studies where weaker tide intensity contributed to increased stratification in the PRE (e.g., Pan et al. (2020)).

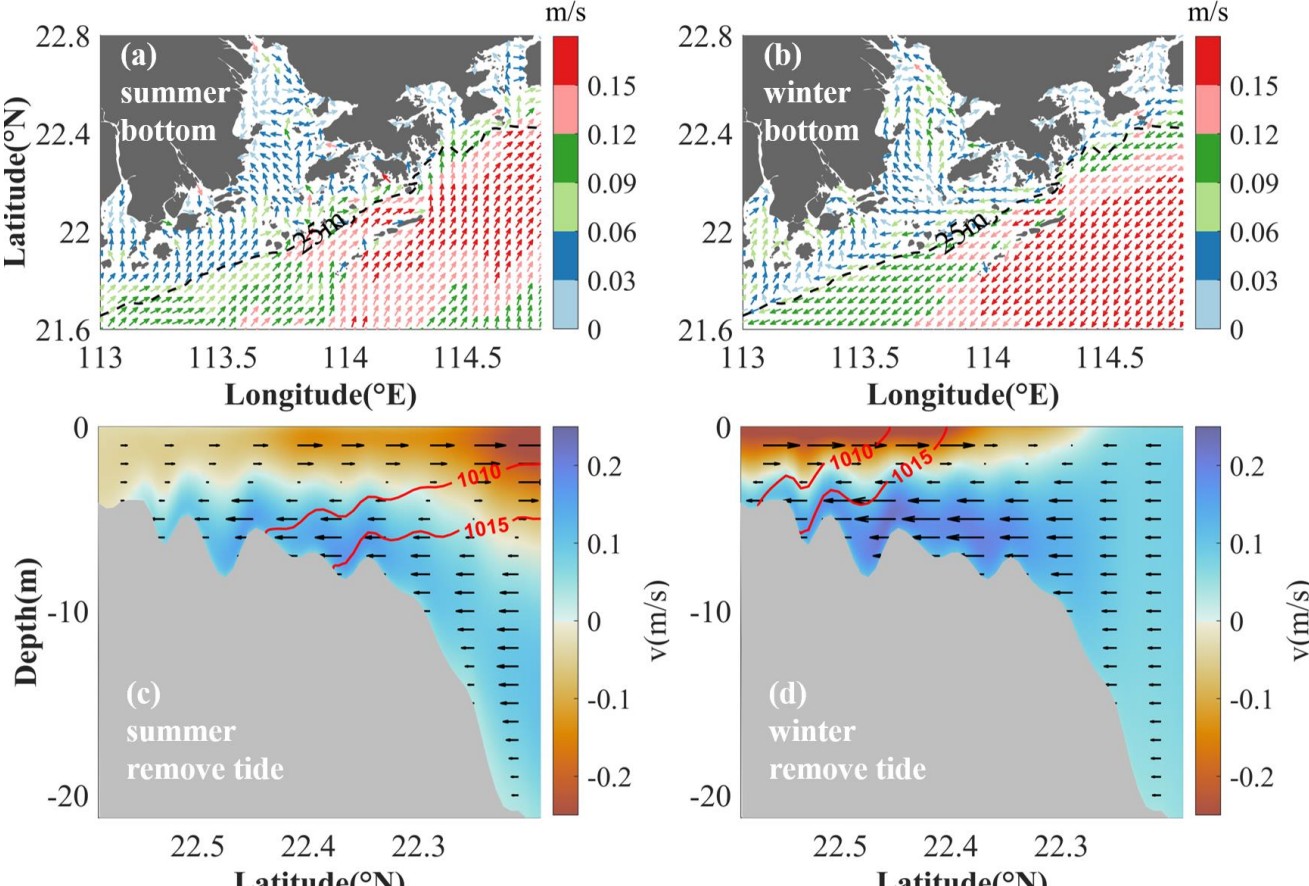

**Figure 9: The flow field of removed tidal currents at the bottom layer in summer (a) and winter (b), respectively. The color bar represents the magnitude of velocity. (c-d) Without the tidal currents, the along transect velocity (color and arrow, positive indicate the onshore intrusion) and density contour of 1010 kg·m⁻³ and 1015 kg·m⁻³ (red lines) in AB during summer and winter, respectively.**

Consequently, the removal of tidal forces intensified particle accumulation in the middle of the WEST and northeastern parts of the EAST region during summer (Fig. 10a). Conversely, in the MO and HK regions, the absence of tides generally decreased the likelihood of accumulation, particularly in MO waters. In winter, the intensified onshore intrusion led to higher accumulation in the UPPER regions (Fig. 10b). In contrast, in the WEST region, tides significantly hinder particles from exiting the estuary because of the intensified landward motion of the westward transport shelf current, leading to a negative accumulation anomaly when the tides are removed. Thus, the tidal current mainly resists bottom intrusion water in the midwestern estuary by changing the mixing intensity and density structure.

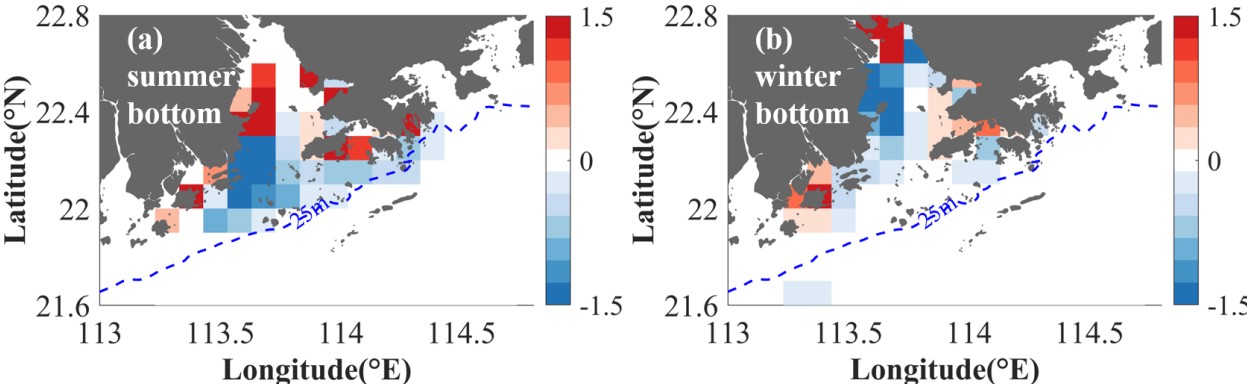

**Figure 10: Probability anomaly in the removing tide current case during summer (a) and winter (b). Anomaly is used to calculate the results of reduced river discharges to subtract the results in the standard case. A negative value represents the strengthened offshore transport without tidal current.**

Same as the standard case, there is a negative correlation between the $\nabla_h \vec{V}_h$ and the water accumulation (Fig. 11a, b). The arrow represents the changes of $\nabla_h \vec{V}_h$ and accumulation due to the removal of tidal currents in the region, with significant changes in the accumulation. The decrease/increase in summer accumulation probability in the MO/EAST region was associated with the weakening or strengthening of the net convergence of the current due to the removal of the tidal current. Similarly, changes in winter accumulation in the UPPER, WEST, and EAST regions were also related to current divergence. However, it is also noted that the WEST and MO regions where have relatively strong $\nabla_h \vec{V}_h$, the changes in accumulation are not significantly influenced by this divergence.

The impact of tides on water transport among the various identified subregions was investigated, as shown in Fig. 11c-d. In summer, removing the tide resulted in strengthened accumulation in the EAST and UPPER regions. The WEST region continued to exhibit a strong converging trend owing to the strengthened landward intrusion current, accumulating particles across the estuary. Notably, in the absence of tides, the MO region no longer converges with particles from other regions. Furthermore, particles originating from the EAST and UPPER regions ceased reaching the HK and MO regions. During winter, the UPPER region displays a heightened attraction to particles from the WEST, EAST, and MO regions, which is associated with intensified onshore currents from the lower estuary. The converging trend in the western region diminished rapidly, reducing its role as a significant trap for water from other regions. There were no significant changes in EAST and HK; only MO trapped more water from its own regions.

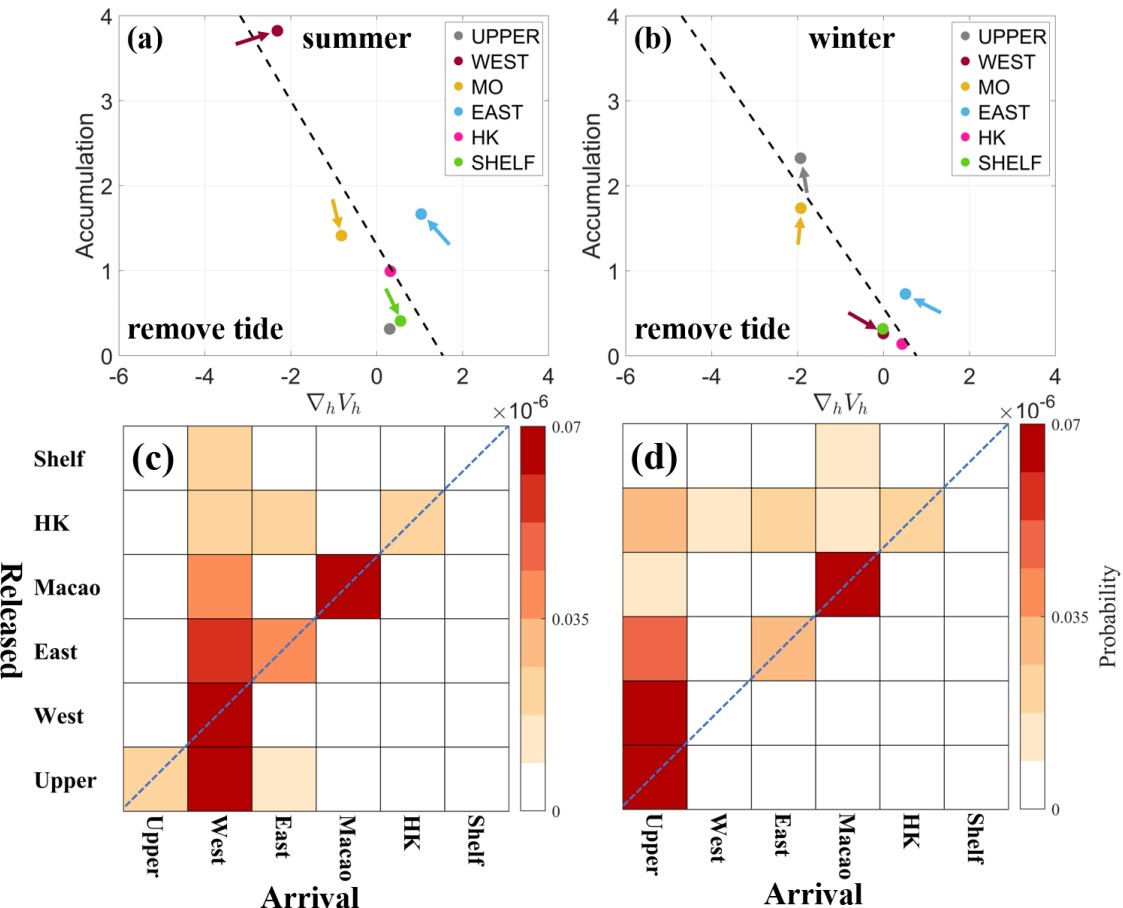

Figure 11: (a-b) Scatter plot of accumulation probability against $\nabla_h \vec{V}_h$ for various subregions during summer and winter in removing tide cases. The arrow represents the changes of $\nabla_h \vec{V}_h$ and accumulation due to the removing of tide in each subregion. (c-d) The connection between six regions for removal of tidal current at the bottom layer during summer and winter time, respectively.

## 4.2 River discharge

Based on the circulation pattern of reduced river discharge, a significant decrease in seaward currents was observed in the entire estuary during both summer and winter (Fig. 12a, b). Along the AB transect, the landward velocity increases in the upper estuary, and seaward motion is accelerated in the lower estuary during winter (Fig. 12d), which is associated with the divergence and convergence of the accumulation probability in the middle/lower estuary in winter (Fig. 13b). Increased landward currents have led to accumulation in the UPPER region. In summer, reduced stratification leads to a well-mixed water column in the inner estuary, and the landward movement of density to the middle estuary blocks the water in the upper part of the estuary (Fig. 12c). As a result, accumulation in the inner estuary increased in summer when river discharge was reduced (Fig.13b).

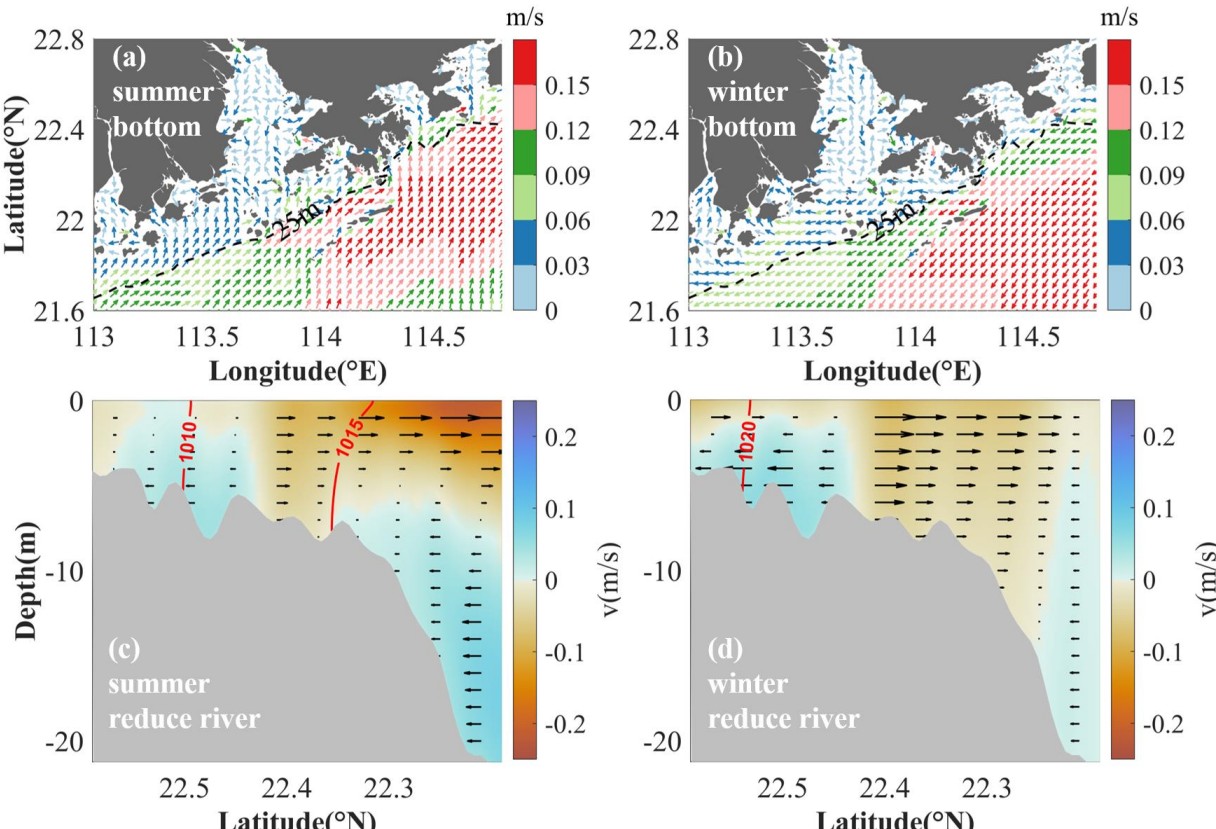

**Figure 12: The flow field of reduced river discharges at the bottom layer in summer (a) and winter (b), respectively. The color bar represents the magnitude of velocity. (c-d) With fewer river discharges, the along transect velocity (color and arrow, positive indicate the onshore intrusion) and density contour of 1010 kg·m⁻³ and 1015 kg·m⁻³ (red lines) in AB during summer and winter, respectively.**

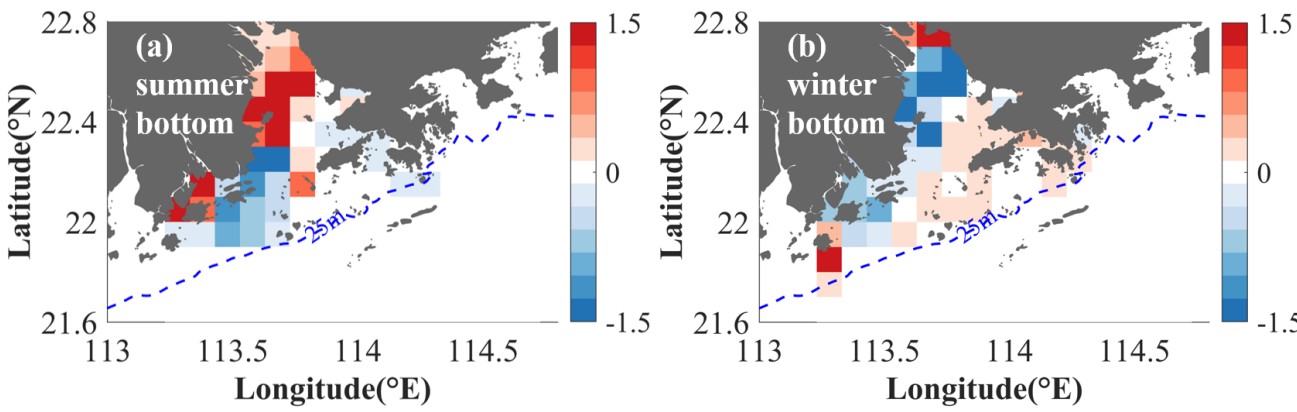

**Figure 13: Probability anomaly in the reducing river discharge case during summer (a) and winter (b). Anomaly is used to calculate the results of reduced river discharges to subtract the results in the standard case. The negative value represents the strengthened offshore transport with fewer river discharges.**

Generally, major changes in accumulation are governed by the current convergence. In both seasons, variations in the net

velocity convergence resulted in corresponding changes in particle accumulation in the UPPER, WEST, and EAST regions (Fig. 14a, b). Increased accumulation is accompanied by decreased net divergence.

Similarly, Fig. 14c-d shows that with reduced river discharge, the transport connections between the UPPER, WEST, and
MO regions were greatly affected. In summer, for the UPPER regions, the converging ability becomes stronger because of the rapidly reduced offshore currents and the blocking of the landward density front; more particles from the UPPER remain in the origin region and no longer move to the EAST and MO regions. Furthermore, reduced river discharge resulted in fewer particles from the MO region accumulating in the WEST region. Instead, the EAST regions contributed more to accumulation in the WEST regions. During winter, the interconnections among the six subregions were similar to those in
the standard case (Fig. 6b). It should also be noted that the WEST regions converge fewer particles than the EAST and HK regions. The MO regions attracted particles from the WEST regions instead of from the SHELF.

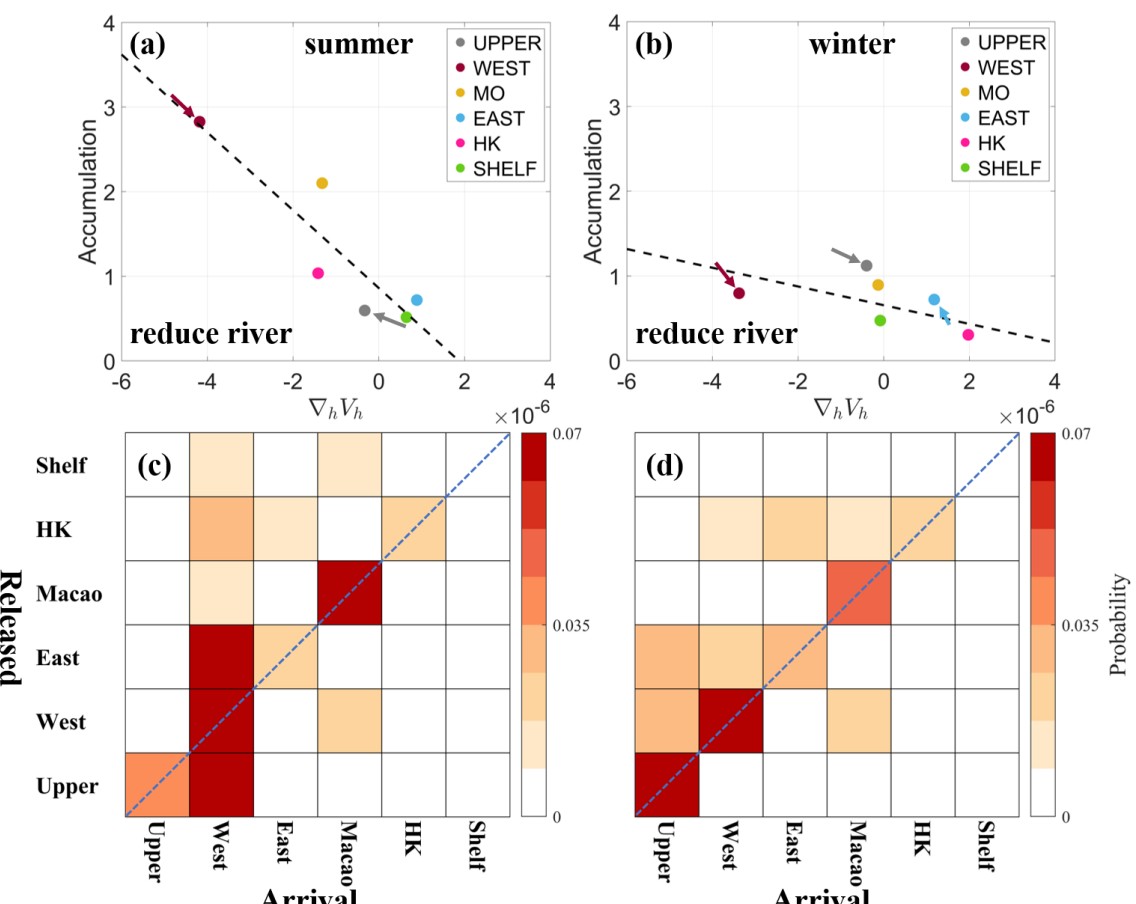

**Figure 14: (a-b) Scatter plot of accumulation probability against $\nabla_h \vec{V}_h$ for various subregions during summer and winter in river-**
**reduced cases. The arrow represents the changes of $\nabla_h \vec{V}_h$ and accumulation due to the reduction of discharge in each subregion. (e-f) The connection between six regions for the case with reduced river discharge.**

**5 Conclusions**

In this study, the Lagrangian method and Markov Chains were applied to illustrate the accumulation trends in different

PRE regions during typical monsoon seasons.

The accumulation probabilities were obtained from the Markov Chains. Generally, surface offshore transport is always

quicker than that at the bottom owing to the strong offshore current and river discharge, which are related to the relatively

small accumulation at the surface layer. At the bottom, a high accumulation of water appears in the lower estuary in summer

and moves shoreward in winter owing to the reduced river discharge and intensified density front. Based on these

accumulation patterns, we identified six subregions in the PRE: UPPER, WEST, EAST, MO, HK, and SHELF. Across the

subregions, there is a negative correlation between the net divergence ($\nabla_h \vec{V}_h$) and the accumulation probability, the

intensified negative $\nabla_h \vec{V}_h$ provide the favorable conditions for water accumulation. The connections between the six

subregions are discussed to illustrate the transport structure in the PRE. During summer, WEST and MO, with substantial

net negative divergence and strong fronts, are powerful accumulation targets that attract particles from almost the entire

estuary. The EAST and HK waters show a westward motion and are transported to the western estuary. In winter, the

accumulated regions showed self-correlations, and particles were more likely to remain in the original regions. The UPPER

becomes a major accumulation region owing to the blocking of the density front and the largely decreased river discharge.

HK waters are transported to almost the entire estuary, contributing mainly to the accumulation in the WEST regions under

westward currents.

Sensitivity experiments were conducted to evaluate the effects of tidal currents and river discharge on accumulation patterns.

Generally, tidal currents and river-induced gravitational circulation affect accumulation in different ways and affect

different regions of the estuary. Their joint effects controlled the accumulation pattern. Tide currents promote accumulation

in the WEST and UPPER regions during winter and in the MO and HK regions during summer through increased density

stratification and changes in water column mixing (Fig. 15). Increased river discharge is conducive to seaside transport in

the UPPER and WEST regions during summer and in the HK region during winter, which is related to the intensified

offshore current and seaward movement of the density. With the removal of tidal currents and reduced river discharges, the

intensified landward current and westward transport current from HK waters and the adjacent shelf will accelerate bottom-

water intrusion from the lower estuary into the upper estuary.

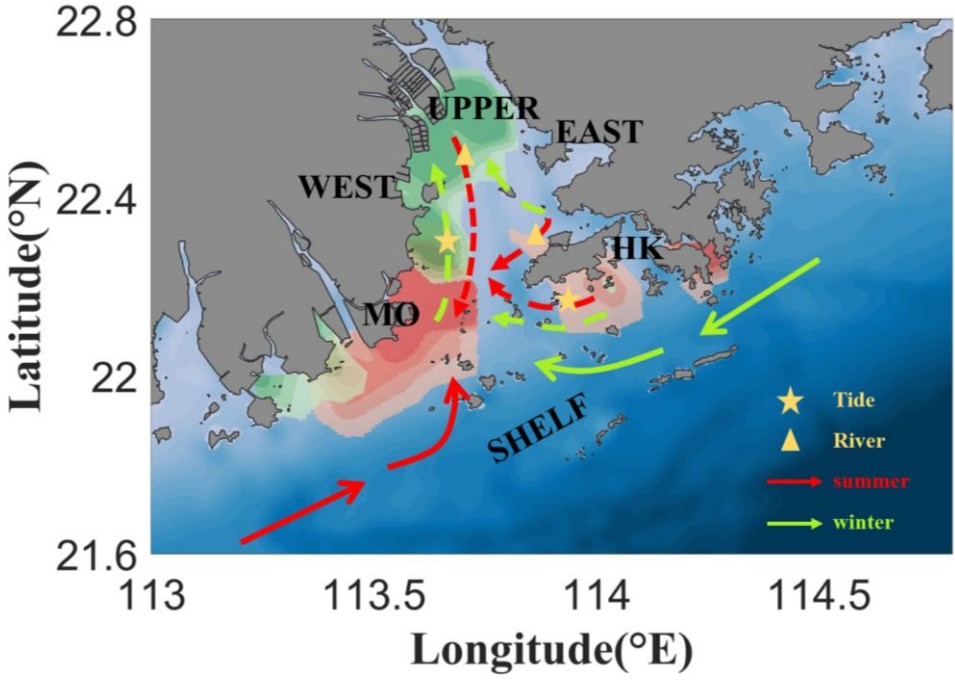


**Figure 15: The accumulation connections schematic in the PRE during summer (red arrow) and winter time (green arrow). The map color in red represented the high accumulation probabilities in summer, while green represented winter. The star indicated that the tide dominated the current, and the triangle represented river discharge.**

**Data availability**

The corresponding authors can provide all raw data upon request.

**Author contributions**

M. L. conducted the investigation, methodology, and writing – original draft preparation; S. A. conducted the methodology and writing – review and editing; Z. C. conducted the conceptualization, supervision, and writing – review and editing; and T.Z. conducted the conceptualization and writing – review and editing.

**Competing interests**

The authors declare that they have no conflict of interest.

**Acknowledgements**

This work was supported by CORE, which is a joint research center for ocean research between the Laoshan Laboratory and HKUST. The simulation was performed at the SICC, supported by SKL-IOTSC, University of Macau.

**Financial support**

This work was funded by the Science and Technology Development Fund, Macau SAR (0093/2020/A2, 001/2024/SKL), and the Open Research Project Program of the State Key Laboratory of the Internet of Things for Smart City (University of Macau) (Ref. No.: SKL-IoTSC(UM)-2021-2023/ORPF/A20/2022), National Natural Science Foundation of China (NSFC) under Project (42076026), Independent Research Project Program of the State Key Laboratory of Tropical Oceanography (LTOZZ2102), and General Research Fund (project ID 15216422) from the Research Grants Council of Hong Kong.

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
