# Peer review of "Exploring Water Accumulation Dynamics in the Pearl River Estuary from a Lagrangian Perspective"

_EGUsphere, 2024_

## Author Comment (AC1)

This paper investigates seasonal patterns of water particles accumulation in Pearl River Estuary subregions following a Lagrangian approach. The authors determined flow fields using an Eulerian model and then monitored particles trajectories for 30 days in the summer and 30 days in the winter by implementing a3D Lagrangian model. Their main conclusion is that plume fronts and velocity convergence are important factors in the determination of water accumulation dynamics while they also identify a negative correlation between divergence and accumulation probability. I reckon that the manuscript contains work that is worthy to be published but it also needs considerable improvements before this happens. Main concerns have to do with phrasing, figures quality and inadequate information supplied. I provide the following general and specific comments which hopefully the authors will find useful and constructive.

**Response:** Thanks for your insightful suggestions and comments, which help to improve our work. We carefully revised our manuscript based on your suggestions and listed our responses as follow.

**General comments**

1. The manuscript needs a thorough revision in terms of language. Specifically, the use of present tense throughout the manuscript is recommended instead of past tense.
   Response: Thanks for your suggestions. We made a thorough language editing in the revised manuscript. And we use present tense instead of past tense in the modified manuscript.

[Figure]

2. A more detailed description of the PRE dynamics is demanded so that the reader can appreciate the unique and complex physical processes in this ecosystem. For instance, the fact that the river discharge in the PRE is higher in the summer and lower in the winter is counterintuitive as in most systems, discharges are higher during winter due to increased rainfall and runoff. It is therefore recommended to the authors to mention this specifically in their introduction. In addition, the monsoonal cycles need to be mentioned as they are important for the water circulation. Relevant information could be included in a separate paragraph either at the end of the Introduction or at the start of section 2.
   **Response:** Thanks for the suggestion. In the revised manuscript, we provide a more detailed description of the PRE dynamics in Line 26-47 in introduction, including the tidal currents, river discharge and wind forcing.

The Pearl River Estuary (PRE), located in the northern South China Sea (NSCS) (Fig. 1a), is influenced by the East Asian Monsoon, with northeasterly winds prevailing in winter and southwesterly winds prevailing in summer (T. Li & Li, 2018). Thus, in the PRE, winter is characterized as a dry season, and summer is characterized as a wet season due to the large rainfall induced by the moist air brought from the South China Sea; consequently, the river discharge in summer (~ 20,000 $m^3s^{-1}$) is approximately five times more than that in winter (~ 3,600 $m^3s^{-1}$) (Harrison, Yin, Lee, Gan, & Liu, 2008). This is quite different from many other river deltas, such as the Mississippi deltas, where river discharge reaches a maximum in winter and spring, but is reduced in summer and autumn (Lane et al., 2007).

As a bell-shaped estuary, the width increases from approximately 5 km at the upper end to 35 km at the lower end. Despite the two narrow, deeper channels (~20 m in depth), the PRE is shallow, with a water depth of approximately 2-10 m. The PRE is a partially mixed estuary in which circulation is jointly controlled by river discharge, tides, wind, and topography (Ascione Kenov, Garcia, & Neves, 2012; Banas & Hickey, 2005; Gong, Shen, & Hong, 2009; C. He, Yin, Stocchino, & Wai, 2023; C. He, Yin, Stocchino, Wai, & Li, 2022; Liu, Zu, & Gan, 2020). There are two distinct dynamic regimes in the PRE. The narrow upper part of the PRE shows classical gravitational circulation, whereas in the wider lower part of the PRE, where the Coriolis effect becomes essential, the topography and interaction with the monsoon-driven shelf current complicate the circulation (Dong, Su, Ah Wong, Cao, & Chen, 2004; Wong et al., 2003; Zu & Gan, 2015). Gravitational circulation occurs in the two deep channel regions, whereas currents show precise seasonal characteristics over the shallower western estuary. Geostrophic wind-driven coastal currents intrude into the PRE during the summer upwelling season (Zu & Gan, 2015), whereas seaward buoyancy-driven coastal currents flow out of the PRE during winter(Dong et al., 2004; Wong et al., 2003). The alternation of the spring-neap tide and variation in river discharge play crucial roles in modulating stratification and mixing inside the PRE (Mao, Shi, Yin, Gan, & Qi, 2004; Pan, Lai, & Thomas Devlin, 2020; Zu, Wang, Gan, & Guan, 2014). Strong tidal mixing in the middle PRE has led to the conversion of estuarine river plumes into buoyancy-driven coastal currents (Dong et al., 2004; Zu et al., 2014).

3. A deeper insight is demanded when discussing the results, particularly on the inter-play among the three basic physical mechanisms of gravitational circulation, tidal and shelf currents in the estuary. It is generally not a good practice to merge results with discussion. It is recommended to detach discussion from section 3 and put it in a separate paragraph. In this way, the core messages of this work can be better illustrated.

**Response:** Thank you for the comment. In the revised manuscript, we have separated the Discussion from the Results. Following the reviewer's suggestion, changes in circulation and density structure were examined to gain a deeper understanding of the influences of river-induced gravitational circulation, tidal effects, and shelf currents.

As the tidal current affects the intensity of mixing in the water column, once removed, the offshore motion in the UPPER region is reduced in both summer and winter. During winter, the intrusion water from the shelf moves straight landward and arrives at the upper part of the estuary. Along the transect of AB (Figure R1a), we further checked the changes in the density stratification and vertical structure of circulation (Figure R2). The intensification and landward movement of the bottom

intrusion is associated with the stronger density stratification. The removal of the tidal current leads to weaker mixing and intensified stratification, as also revealed in previous studies where weaker tide intensity contributed to increased stratification in the PRE (e.g., Pan et al. (2020)).

Consequently, when the tidal currents were removed in summer, the accumulation probability in the midwestern region increased (Figure 7a in the original manuscript), whereas, during winter, the intensified onshore intrusion led to a higher accumulation in the UPPER region (Figure 7b in the original manuscript). Thus, the tidal current mainly resists bottom intrusion water in the midwestern estuary by changing the mixing intensity and density structure.

Similarly, we checked the changes in bottom intrusion and density stratification in the river reduction case. With reduced river discharge, the seaward motion in the entire estuary primarily decreased in both winter and summer (Figure R3). Along the AB transect, the landward velocity increases in the upper estuary, and seaward motions are accelerated in the lower estuary during winter (Figure R3d), which is associated with the negative/positive value of the accumulation probability in the middle/lower estuary during winter (Figure 8b in the original manuscript). In summer, reduced stratification leads to a well-mixed water column in the inner estuary, and the landward movement of density to the middle estuary blocks the water in the upper part of the estuary (Figure R3c). As a result, accumulation in the inner estuary increased in summer when river discharge was reduced (Figure 8a in the original manuscript).

Generally, tidal currents and river-induced gravitational circulation affect accumulation in different ways and affect different regions of the estuary. Their joint effects controlled the accumulation pattern. The above explanation was added in the revised manuscript.

[Figure]

Figure R1. (a-b) The flow field of the standard case at the bottom layer during summer and winter time, respectively. (c-d) are the same as (a-b), except for removing the tidal currents. The colorbar represents the velocity of the current. AB indicates the location of the transect to show the vertical structure.

[Figure]

Figure R2. (a-b) The along transect velocity (color and arrow, positive indicate the onshore intrusion) and density contour of 1010 kg/m³ and 1015 kg/m³ (red lines) in AB during summer and winter. The location of AB shown in Figure R.1 (c-d) the same as (a-b), except for removing the tidal currents.

[Figure]

Figure R3: (a-b) The flow field of the river reduced case at the bottom layer during summer and winter time,

respectively. (c-d) The along transect velocity (color and arrow, positive indicate the onshore intrusion) and density contour of 1010 kg/m³ and 1015 kg/m³ (red lines) in AB during summer and winter. The location of AB is shown in Figure R1.

4.  The results analysis and discussion focus mainly on the influence of river discharge, tide, and plume fronts. But the monsoonal cycles and consequently wind forcing, downwelling, upwelling etc. are also important parameters that affect water circulation and accumulation. These are only briefly mentioned in page 6 (lines 140-142) and somehow their effect is understated in the manuscript. The contribution of the monsoonal climate can be reevaluated and better highlighted.

    **Response:** Following the reviewer's suggestion, an additional experiment was conducted to discuss the impact of wind forcing on circulation patterns. In addition, a deeper analysis was conducted to illustrate better the joint effects of river discharge, wind forcing, and tidal currents on the accumulation pattern in the bottom layer.

    Generally, the wind has less influence on the circulation in the estuary (Figure R4). In summer, compared with the river and tide, the wind does not have a significant impact on circulation in the estuary. Only the shelf current outside the estuary was weakened (Figure R4a, c). In winter, the wind mainly affects the current in the lower estuary and adjacent shelf, where offshore motions are reduced, and landward transport to the south of the WEST region is strengthened (Figure R4b, c).

    Consequently, the accumulation pattern and connection among subregions remain the similar to standard case in summer. In winter, higher accumulations occurred mainly along the west side of the estuary (Figure R5b). The strengthened westward current and reduced offshore motion transport more water from Hong Kong and Shelf towards the WEST region (Figure R5d). The connection between other regions is similar to standard case, mainly differences appear in the WEST region.

    By comparing the accumulation patterns and circulation under different dynamic factors, we summarized the relationships among the three controlled factors. In summer, the circulation and accumulation in the estuary are mainly controlled by river and tidal currents, which together contribute to accelerating the offshore motion in most regions. Compared with tidal currents and river discharges, the wind has less impact. In winter, the existence of tidal currents and river discharge leads to the convergence of water parcels in the WEST region and divergence in the UPPER region. Wind mainly affected the shelf current and accelerated the offshore transport on the western side of the lower estuary.

    Since the effect of the wind in the estuary is relatively small, we did not put this additional case in the manuscript. In the revised manuscript, we clarify the influence of wind forcing:
    Line 267-269: "The tides and rivers had essential influences on the estuarine circulation and associated mass transport in the PRE, whereas the wind mainly affected the shelf current and had less influence on the mass accumulation inside the estuary (figures not shown here)."

[Figure]

Figure R4. The flow field of reduced wind forcing at the bottom layer in summer (a) and winter (b), respectively. The flow field of reduced wind forcing in transect AB at the bottom layer in summer (c) and winter (d), respectively. The colorbar represents the onshore velocity, and the positive value means the landward motion in transect AB. The arrow represents the current direction, and the length represents the strength of the currents. The red line represents the isopycnal of 1010 kg/m$^3$ and 1015 kg/m$^3$.

[Figure]

Figure R5. Accumulation probability at the bottom layer during summer (a) and winter (b). The colorbar indicate the magnitude of accumulation probability. (c-d) The connection between the six subregions is when the wind force is reduced during the summer and winter, respectively. The horizontal and vertical axes represent the arrival and release regions, respectively.

5. FontSizes and Figures sizes need to be enlarged.
   **Response:** FontSizes and Figures sizes has been enlarged in the revised manuscript.

**Specific comments**

**Introduction:**

Lines 43-46. It would be useful to expand and elaborate on why it is important to know and identify areas of water accumulation. Is this interesting only for the PRE or for any estuary? Is eutrophication the only problem or is it particularly concerning for the PRE? Note also that eutrophication is not an anthropogenic activity but one of its adverse consequences.

The introduction can be enriched with some further literature on the topic. Is this a topic that has been explored before and to what extent?

**Response:** Thank you for the comment. The estuary is a transition region between the land and ocean and is facing increasing pressure for water protection (Callahan et al., 2004) due to recent intensified human activities. It should be noted that under complicated water motions, water is more likely to be trapped in some regions. For example, the salt wedge provides a sink for pollutants in estuaries (Vermeiren, Muñoz, & Ikejima, 2016), and higher concentrations of microplastics have been observed at the salt wedge in the Rio de la Plata estuary (Acha et al., 2003). The accumulation of fine particles in some estuarine regions with weak flushing traps had high concentrations of heavy metals in them (Balachandran et al., 2005). Therefore, exploring the accumulation dynamics in estuaries provides favorable conditions for revealing the formation of pollutant sinks (Mestres et al., 2006; Tao, Niu, Dong, Fu, & Lou, 2021; Vermeiren et al., 2016; A.-j. Wang et al., 2016).

Eutrophication is not the only concern of PRE. D. Zhang et al. (2013) found that trace elements have the highest concentrations in the western PRE. Microplastics are concentrated in Hong Kong water and the western part of the PRE owing to the influence of tidal currents and the Pearl River outflow (Lam et al., 2020). Tao et al. (2021) revealed that the accumulation of nitrogen and silicate occurred in the upper part of the PRE, and phosphorus pollution was observed in the northeast of the PRE (e.g., Shenzhen Bay). These phenomena are also related to the accumulation patterns in the PRE.

In the previous investigation, the phenomenon of accumulation of pollutants/materials in particular regions were noticed, but do not have a clear understanding of its spatial pattern and underlying physical control. In this study, using the PRE as a typical example, we attempted to determine the connection between different subregions in the PRE to verify the physical processes controlling accumulation patterns.

Following the reviewer's suggestions, we integrate the above information in the revised manuscript:

Line 53-71:

In estuaries, some regions are more likely to attract water because of complicated current circulation,

which can be considered as stronger horizontal convergence targets for some materials (T. Wang et al., 2022). For example, the salt wedge acts as a significant pollutant sink in an estuary, and a higher concentration of microplastics is always obtained at the salt wedge in the Rio de la Plata estuary (Acha et al., 2003; Vermeiren, Muñoz, & Ikejima, 2016). Areas accompanied with higher concentration of nitrogen and phosphorus are usually appearing eutrophic (Tao, Niu, Dong, Fu, & Lou, 2021). Heavy metal pollution in estuaries has been observed in areas that prefer to concentrate fine particles (Balachandran et al., 2005). Therefore, identifying the accumulation areas in estuary-shelf systems provides an adequate estimate for surveying pollutant sinks(Mestres et al., 2006; Tao et al., 2021; Vermeiren et al., 2016; A.-j. Wang et al., 2016).

With intensified human activities, pollutant sinks related to the accumulation phenomena in the PRE have attracted attention. Tao et al. (2021) revealed that the upper part of the PRE is a target sink for nitrogen and silicate. D. Zhang et al. (2013) found that trace elements prefer to accumulate on the PRE's west side. Higher concentrations of microplastics have been observed in western estuaries and Hong Kong waters (Lam et al., 2020). Similarly, studies on hypoxia have shown that the convergence of buoyancy-driven currents and wind-driven shelf flows contributes to the formation of stable water columns, providing favorable conditions for the development of hypoxic zones (e.g., D. Li et al. (2021); X. Li et al. (2020)). The high frequency of hypoxia in the estuary during summer is related to the strong stratification of the water column (Y. Cui, Wu, Ren, & Xu, 2019; H. Zhang & Li, 2010). These accumulation patterns in the PRE are more concerned with the measurement of pollutant concentrations (Tao et al., 2021), estimation of the pollutant accumulation rate (L. Zhang et al., 2009), and discussion of the sources of pollutants (Ye, Huang, Zhang, Tian, & Zeng, 2012), and lack a discussion on the understanding of accumulation spatial patterns and underlying physical control.

**Methods:**

Model settings are either missing or insufficiently presented. Where are the ROMS model's boundaries set? Which area do they cover? A figure with the model grid and bathymetry should be added. How much is the river discharge in the summer and how much in the winter? Please provide a sufficient summary of the model setup.

**Response:**

Thanks for your suggestions. Our PRE model is developed using Regional Ocean Model System (ROMs) (Shchepetkin & McWilliams, 2005). The model region covers estuary and adjacent shelf between 112.3°E-115.68°E and 20.89°N-23.13°N (Figure R9). We adopt an orthogonal grid and the resolution increases from ~1km over shelf to ~200 m inside the estuary. In vertical direction, we use the terrain-following s-ordinate (Song & Haidvogel, 1994) to discretize the water column into 30 levels. The monthly averaged river discharge data is obtained from Ministry of Water Resources of China. During summer and winter, the river discharge is approximately 30000 $m^3$/s and 10000 $m^3$/s, respectively. Wind forcing, heat flux, and precipitation are obtained from the ERA5 atmospheric reanalysis data by European Centre for Medium-Range Weather Forecasts (ECMWF), and they are used to force ocean circulation through the implementation of the bulk computation algorithm (Fairall, Bradley, Hare, Grachev, & Edson, 2003). The shelf current is obtained from a coarser model (Deng et al., 2022), which can cover the North South China Sea and provide the information of barotropic and baroclinic velocities, temperature, salinity, sea level along the boundaries of PRE model. Turbulent and diffusion are determined by Mellor-Yamada

2.5 turbulence-closure module (Mellor & Yamada, 1982). Eight major tidal harmonic constants ($M_2$, $S_2$, $K_2$, $N_2$, $K_1$, $O_1$, $P_1$, $Q_1$ ) as well as the $M_4$ obtained from Zu, Gan, and Erofeeva (2008) are included to calculated the tidal induced current and elevation along the boundaries.

Following the reviewer's suggestions, we have integrated the above information in the revised manuscript and added Supplementary Figure R7 to introduce the seasonal river discharge and wind direction.

[Figure]

Figure R6 The bathymetry of the study area. The black dotted and dashed lines represent the isobath of 25 m and 50 m. The yellow line defines the seaside boundary of the PRE. LDY, MO, HK, and NSCS represent Lingdingyang, Macau, Hong Kong, and the northern South China Sea, respectively. The red line represents the model boundary.

[Figure]

Figure R7. (a-b) The seasonal mean wind speed (m/s) during summer (June–August) and winter (December–February), respectively. (c) The monthly river discharge in the PRE.

Line 78-79 'Data of shelf currents were obtained by the coarser – resolution simulation'. Is there a coarser model also used? Please explain.

**Response:** Thank you for the comment. A well-validated coarse model simulated the shelf current covering the entire NSCS. The results provided the barotropic and baroclinic velocities, temperature, salinity, and sea level along the boundaries of the PRE. This model has been validated and used in previous investigations (L. Cui, Cai, & Liu, 2023; L. Cui, Liu, Chen, & Cai, 2024; Deng et al., 2022).

[Figure]

Figure R8. The domain of coarser model.

Line 88 Has the model been validated?

**Response:** Yes, although this model was established using climatological forcing to obtain the main features of circulation, we carefully validated the simulation of the hydrodynamic properties against data from both satellite remote sensing and long-term observations. The seasonal sea surface temperature from remote sensing and the simulation showed that the simulation captured the basic distribution and magnitude of SST (Figure R9). Considering the uncertainties in coastal salinity from remote sensing, long-term (2000-2019) observed hydrodynamic data at the SM19 station near Hong Kong were used to validate that the model accurately captured the seasonal variability of the hydrodynamic features (Figure R10).

For the basic circulation pattern, the simulation accurately reproduced the major features. In summer, an eastward shelf current established over the shelf of the Pearl River Estuary, with the onshore invasion of colder (and saltier) shelf water, is extensively intensified in the coastal seas east of the PRE (Figure 3a, b in the original manuscript). In winter, when the stronger monsoonal northeasterly wind prevailed over the study area, the coastal waters east of the PRE became much colder and mixed much better in the water column (Figure 3c, d in the original manuscript). These features are similar to those of previous investigations in this area (e.g., Cai, Liu, Liu, and Gan (2022); Z. Liu et al. (2020)), and the results from this model were successfully used in previous investigations, which provided indirect support for the

quality of the results.

In the revised manuscript, we have explained the model validation and put Figure R9, 10 in the Supplementary Material.

Line 118-121: "This model, primarily based on climatological data, was carefully verified using satellite remote sensing and long-term observations to ensure an accurate representation of the hydrodynamic properties (Fig. S2, 3). Generally, the model captures the seasonal features of circulation in this region and has been used in previous studies (Cai et al., 2022; Chu et al., 2022a; L. Cui et al., 2024)."

[Figure]

Figure R9. (a-b) Climatological SST anomaly during summer and winter from MUR SST reanalysis product from the Jet Propulsion Laboratory (JPL) of NASA (2002-2021). (c-d) are the same as (a-b) but for the model results.

[Figure]

Figure R10. Time series of the surface simulated and observed potential water densities at the SM19 station near Hong Kong (https://cd.epic.epd.gov.hk/EPICRIVER/marine/?lang=zh_cn). The observed data at SM 19 from 2000-2019 was used to obtain the climatological monthly mean density, and the error bar indicates the density of STDs for each month in 20 years. The location of SM19 is shown in Figure R9a.

Line 90 Please state the implemented diffusion coefficients.
**Response:** The diffusion coefficient was calculated using the hydrodynamic model through the Mellor-Yamada 2.5 turbulence-closure scheme (Mellor & Yamada, 1982), which has been widely used in coastal simulations (Choi, Park, Choi, Jung, & Kim, 2021; J. Liu, Lu, & Li, 2019; Robertson & Hartlipp, 2017).

The diffusion coefficient was determined by solving the prognostic equations for the turbulent kinetic energy, turbulent kinetic energy dissipation rate, and turbulent length scale, which vary spatially and temporally within the study area.

We clarify this in the revised manuscript:

Line 104-106: "Vertical turbulence and diffusion coefficient are determined by the Mellor-Yamada 2.5 turbulence-closure module (Mellor & Yamada, 1982), which provides the turbulent mixing coefficient."

Line 100 It is understood from Figure 3 that the calculations are in 3D, what is the vertical dimension of the grids?

**Response:** In our model calculations, we used the terrain-following s-ordinate (Song & Haidvogel, 1994) to discretize the water column into 30 levels, and a higher resolution for both the surface and bottom boundary layers was designed to capture the motions better there. Using the calculated 3D velocities from the hydrodynamical model, we explored accumulation dynamics in both the surface and bottom layers of the PRE.

In the revised manuscript, we clarify the vertical dimension of the grid:

Line96-97: "In the vertical direction, we used the terrain-following S-ordinate (Song & Haidvogel, 1994) to discretize the water column into 30 levels."

**Results and Discussion:**

Line 145-146 Hypoxia can be common in microtidal estuaries such as the PRE or the Mississippi (see for example Schiller et al.2011) and is usually associated with stratification induced by river flows in the absence of significant mixing. It would be interesting to see the density at the surface in Figure 6 for both summer and winter in addition to the bottom ones. I presume this would show strong stratification during the summer.

**Response:** Following the reviewer's suggestion, we plotted the surface density during both summer and winter. As mentioned by the reviewer, during summer, stratification occurred over the estuary with a lower surface density (Figure R11). As reported in previous studies, stratification affects the vertical diffusion of dissolved oxygen, which is a crucial factor in hypoxia (Y. Cui et al., 2019). H. Zhang and Li (2010) revealed that stratification is one of the main motivations for hypoxia in the western shoal and Hong Kong waters in the PRE during the summer.

In winter, under weaker river discharge and a robust northeasterly monsoon, the density difference between the bottom and surface water is relatively small. Stratification was reduced, and the water column was relatively well mixed (Dong et al., 2004).

We explain the stratification during summer-time and its influences on hypoxia:

Line 67-68: "The high frequency of hypoxia in the estuary during summer is related to the strong stratification of the water column (Y. Cui et al., 2019; H. Zhang & Li, 2010)."

[Figure]

Figure R11. The density distribution in the PRE, the unit of density, is kg/m$^3$.

Line 148 and 200-212. The role of the gravitational circulation is a bit obscure from the figures. A comparison between Figure 3 b and d shows that the flow arrows at the bottom do not differ much. In fact, the flow field is very similar. In addition, the density field at the bottom is also very similar between summer and winter if we compare Figure 6 a and b. This is not unusual. See for example the paper by Wong et al. (2003) which also talks about the PRE. They mention that the bottom salinity front between summer and winter is similar which basically agrees with what we see in Figure 6. Figure 3d shows an increase of the accumulation in the upper estuary in the winter compared to the summer (Figure 3b). Regarding the similar density fronts, how certain is it that the gravitational circulation is responsible for this increase? Figure 5b indicates that the increase of accumulation in the upper estuary is caused by convergence (difference between Figure 5 a and b). Could this be related to the decrease of river discharges?

**Response:** Thanks! According to the AB transect in the standard case (Figure R2 a-b), an obviously strengthened landward current was observed at the bottom, which was beneficial for intensifying the bottom intrusion. Summer has much more river discharge than winter in the PRE (Harrison et al., 2008), and a strong outflow pushes the water to the lower estuary and hinders the intrusion of water into the estuary. In winter, less river discharge has a weaker resistance to intrusion water, which causes strengthened landward movement of the bottom water and pushes the isopycnals to move into the inner estuary. Thus, although the front is similar, the gravitational circulation induced by river discharge contributes to changes in water accumulation in the estuary. And the increased accumulation in the upper estuary is also related to the decrease of river discharge.

Line 164 It would be helpful to add in Figure 4 a picture of the original (starting) distribution of the particles.

**Response:** Thank you for the comment. Initially, the particles were located in the release regions; that is,

the arrival region was the same as the release region. According to the Line 141 in the manuscript, the initial distribution $D^{t_0}$ is $D^{t_0} = [1, 1, 1, \cdots 1]$, which represents that the original distributions of the particles are all on the diagonal location of the matrix (Figure R12):

[Figure]

Figure R12. The location of the initial particle location.

Line 180 Do u and v indicate mean over depth velocity or at the bottom layer?
**Response:** The bottom u and v were used to calculate the bottom divergence.

**Role of Tide and River:**

The main message I get from Figure 7 and 8 is a landward displacement of water accumulation when the tides are removed, and the discharge is reduced respectively. What is the cause for this? It is not very clear from the text.
**Response:**
Thank you for your comments. In Figure R2 and Figure R3, we present the flow field of Transect AB with the removed tidal current/reduced river discharges to show the vertical motion of the current. As Figure R2 shows, when the tidal currents were removed, the mixing was reduced and the stratification was intensified (Pan et al., 2020), which was associated with the landward movement of density in both winter and summer. Consequently, the higher landward accumulation probabilities are related to the strengthened bottom intrusion (Figure 7a, b in the original manuscript). In Figure R3, along the estuary, when river discharges were reduced, the weakened offshore velocity and strengthened landward front in the upper estuary were associated with a higher accumulation probability.

Line 229 How is the negative anomaly defined?
**Response:** The anomaly was calculated as the result of sensitivity experiments minus the standard case. The negative anomaly here means that due to the reduction in river discharge/tidal currents, the accumulation was weakened, and offshore transport was strengthened.

**Conclusions:**

The conclusions read more like a summary. It is suggested to rewrite it so that the core messages of this work are better highlighted.

**Response:** Thanks for reviewers' suggestions. We have rewritten the conclusion in the revised manuscript.

Line 364-388:

In this study, the Lagrangian method and Markov Chains were applied to illustrate the accumulation trends in different PRE regions during typical monsoon seasons.

The accumulation probabilities were obtained from the Markov Chains. Generally, surface offshore transport is always quicker than that at the bottom owing to the strong offshore current and river discharge, which are related to the relatively small accumulation at the surface layer. At the bottom, a high accumulation of water appears in the lower estuary in summer and moves shoreward in winter owing to the reduced river discharge and intensified density front. Based on these accumulation patterns, we identified six subregions in the PRE: UPPER, WEST, EAST, MO, HK, and SHELF. Across the subregions, there is a negative correlation between the net divergence ($\nabla_h \vec{V}_h$) and the accumulation probability, the intensified negative $\nabla_h \vec{V}_h$ provide the favorable conditions for water accumulation. The connections between the six subregions are discussed to illustrate the transport structure in the PRE. During summer, WEST and MO, with substantial net negative divergence and strong fronts, are powerful accumulation targets that attract particles from almost the entire estuary. The EAST and HK waters show a westward motion and are transported to the western estuary. In winter, the accumulated regions showed self-correlations, and particles were more likely to remain in the original regions. The UPPER becomes a major accumulation region owing to the blocking of the density front and the largely decreased river discharge. HK waters are transported to almost the entire estuary, contributing mainly to the accumulation in the WEST regions under westward currents.

Sensitivity experiments were conducted to evaluate the effects of tidal currents and river discharge on accumulation patterns. Generally, tidal currents and river-induced gravitational circulation affect accumulation in different ways and affect different regions of the estuary. Their joint effects controlled the accumulation pattern. Tide currents promote accumulation in the WEST and UPPER regions during winter and in the MO and HK regions during summer through increased density stratification and changes in water column mixing (Fig. 15). Increased river discharge is conducive to seaside transport in the UPPER and WEST regions during summer and in the HK region during winter, which is related to the intensified offshore current and seaward movement of the density. With the removal of tidal currents and reduced river discharges, the intensified landward current and westward transport current from HK waters and the adjacent shelf will accelerate bottom-water intrusion from the lower estuary into the upper estuary.

**Figures:**

Figure 1 Put panel names a and b on the left and right image. Change the colour of 50m isobath to something more visible.

**Response:** Thanks for reviewers' suggestion. We have modified Figure 1 in the revised manuscript.

Figure 3. Use a different colorscale for the vectors (e.g., blue), differences can hardly be spotted with this one. Add an arrow size. Same for the colour of the areas' names. Why do you show results at the surface

only in this occasion and nowhere else? Is your research focusing on the bottom or the entire water column?

**Response:** Thanks for reviewers' suggestions! We will separate the information of flow field from Figure 3 in the original manuscript and get the new version of flow field as Figure R13.

Under the strong offshore current, the accumulation probability on the surface is much weaker than bottom layer. The bottom layer has slower offshore motions and most environmental problems such as hypoxia and heavy metal enrichment occurred mainly in the bottom as well. Therefore, we choose the bottom layer to discuss the internal connection and accumulation dynamics between different regions in the PRE, which can show the pattern more clearly. Following discussion on the accumulation dynamics are focused on the bottom layer.

[Figure]

Figure R13: (a–b) The flow field of the standard case at the surface layer and bottom layer during summertime, respectively. (c-d) are the same as (a-b), but for winter time.

Figure 5. Keep the y axis scale equal between a and b for better comparison.

**Response:** We use the same y axis scale here to show the distinction.

Figure 6. Add results for the surface layer.

**Response:** The density gradient of surface layer is shown as follow:

[Figure]

Figure R14. The density front (kg/m⁴) on the surface layer (a) and bottom layer (b) during summer time, while (c-d) are same as (a-b) in winter.

Figure 7. Are these results given for the entire water column or just the bottom? Make y axis equal between c and d.

Figure 8. Same comments as in Figure 7.

**Response:** Thanks for reviewers' suggestions! The results in Figure 7 are focused on the bottom layer. Compared to the bottom layer, particles on surface layer leave the estuary quickly without significant accumulation effects. Thus, the discussion on hydrodynamic control in this paper are mainly concentrated on the bottom layers. We will make y axis scale keep same like Figure R15, 16.

[Figure]

Figure R15: (a-b) Scatter plot of accumulation probability against $\nabla_h \vec{V}_h$ for various subregions during summer and winter in removing tide cases. The arrow represents the changes of $\nabla_h \vec{V}_h$ and accumulation due to the removing of

tide in each subregion. (c-d) The connection between six regions for removal of tidal current at the bottom layer during summer and winter time, respectively.

[Figure]

Figure R16: (a-b) Scatter plot of accumulation probability against $\nabla_h \vec{V}_h$ for various subregions during summer and winter in river-reduced cases. The arrow represents the changes of $\nabla_h \vec{V}_h$ and accumulation due to the reduction of discharge in each subregion. (e-f) The connection between six regions for the case with reduced river discharge.

---

## Author Comment (AC2)

1. This study uses Lagrangian particle tracking and Markov Chains to study the accumulation dynamics of water or passive/conservative mass in the PRE and connectivity among its subregions. Convergence and fronts were identified as major factors for high accumulation probability. The study is based on a validated model as the authors stated, yet more details on the model set up should be provided, particularly the sensitivity runs on modifying tide and river discharges. The identified seasonal accumulation dynamics in different subregions and their connectivity are interesting, yet more implications on the ecosystem in the PRE can be discussed. For example, based on the sensitivity run of reducing river discharge, should we worry about more pollutants will be accumulated in some regions in dry years? I also suggest the authors to carefully check the grammar and revise their writings to improve the clarity and readability of this manuscript. In addition, I am not familiar with the calculation of Markov Chains and I except other reviewers and the topic editors would have better judgements. Please find specific comments below.

**Response:** Thanks for reviewer's perceptive suggestions and comments, which are useful for us to improve our work. We responded to your concerns separately as below:

*The study is based on a validated model as the authors stated, yet more details on the model set up should be provided, particularly the sensitivity runs on modifying tide and river discharges.*

**Response:** Thanks for reviews' reminder. Our PRE model is developed using Regional Ocean Model System (ROMs) (Shchepetkin & McWilliams, 2005). The model region covers estuary and adjacent shelf between 112.3°E-115.68°E and 20.89°N-23.13°N (Figure R9). We adopt an orthogonal grid and the resolution increases from ~1km over shelf to ~200 m inside the estuary. In vertical direction, we use the terrain-following s-ordinate (Song & Haidvogel, 1994) to discretize the water column into 30 levels. The monthly averaged river discharge data is obtained from Ministry of Water Resources of China. During summer and winter, the river discharge is approximately 30000 $m^3$/s and 10000 $m^3$/s, respectively. Wind forcing, heat flux, and precipitation are obtained from the ERA5 atmospheric reanalysis data by European Centre for Medium-Range Weather Forecasts (ECMWF), and they are used to force ocean circulation through the implementation of the bulk computation algorithm (Fairall, Bradley, Hare, Grachev, & Edson, 2003). The shelf current is obtained from a coarser model (Deng et al., 2022), which can cover the North South China Sea and provide the information of barotropic and baroclinic velocities, temperature, salinity, sea level along the boundaries of PRE model. Turbulent and diffusion are determined by Mellor-Yamada 2.5 turbulence-closure module (Mellor & Yamada, 1982). Eight major tidal harmonic constants ($M_2$, $S_2$, $K_2$, $N_2$, $K_1$, $O_1$, $P_1$, $Q_1$ ) as well as the $M_4$ obtained from Zu, Gan, and Erofeeva (2008) are included to calculated the tidal induced current and elevation along the boundaries.

To test the influences of tide current and river discharge on the accumulation dynamics, two sensitivity tests were carried out. One is by removing the tide, i.e. not considering the tidal induced current and elevation along the open boundary conditions, the other one is by reducing the river discharge to 20% of the control run.

In the sensitivity tests, the experiments involve the removal of tide and a reduction of river discharge to 20%.

Above information will be integrated in Line 94-109 in the revised manuscript.

[Figure]

Figure R1. The bathymetry of study area. The black dotted and dashed lines represent the isobath of 25 m and 50 m. The yellow line defines the seaside boundary of the PRE. LDY, MO, HK and NSCS represent Lingdingyang, Macau, Hong Kong and northern South China Sea, respectively. The red line represents the model boundary.

*The identified seasonal accumulation dynamics in different subregions and their connectivity are interesting, yet more implications on the ecosystem in the PRE can be discussed. For example, based on the sensitivity run of reducing river discharge, should we worry about more pollutants will be accumulated in some regions in dry years?*

**Response:** Based on our sensitivity run, we noticed that the accumulations increased in UPPER and northern part of WEST region due to the reduction of river discharge. Similar results were found in the study of X. Zhang et al. (2013), the concentration of particle organic carbon and suspended solid in above mentioned regions in 2010 was much higher than that in 2011 during summertime, while the river discharges in 2010 was only half of the amount in 2011 during that time periods. Therefore, we thought it is possible that more pollutants will be accumulated in some regions in dry years.

*I also suggest the authors to carefully check the grammar and revise their writings to improve the clarity and readability of this manuscript.*

The grammar and the writing were improved in the revised manuscript.

[Figure]

**Certificate of Elsevier
Language Editing Services**

The following article was edited by Elsevier Language Editing Services:

Exploring Water Accumulation Dynamics in the Pearl River
Estuary from a Lagrangian Perspective

**Ordered by:**

Zhongya Cai

Estimated Delivery date:
2024-04-22
Order reference:
ASLEEX1055154

2. Title: suggest revising it to specifically indicate that this study is about the accumulation dynamics of water (or passive material). Add 'a' before 'Lagrangian Perspective'.

   **Response:** Thanks a lot. The titles changes into 'Exploring Water Accumulation Dynamics in the Pearl River Estuary from a Lagrangian Perspective' in the revised manuscript.

3. Abstract and Conclusion need to be more concise. These long paragraphs will distract readers from the main takeaways. It is also necessary to mention that results in this study are based on a model with climatological forcings and not specific to a certain year that may have large variations in the hydrodynamics.

   **Response:** Thanks for reviewers' suggestions. The abstract and conclusion are compressed in the revised manuscript and the study is based on a climatological forcing model are mentioned in the manuscript Line 102-104 and Line 118-121.

   Line 102-104: "The shelf current was obtained from a coarser model with good validation that can cover the North-South China Sea and provide information on the barotropic and baroclinic velocities, temperature, salinity, and sea level along the boundaries of the PRE (Deng et al., 2022)."

   Line 118-121: "This model, primarily based on climatological data, was carefully verified using satellite remote sensing and long-term observations to ensure an accurate representation of the hydrodynamic properties (Fig. S2). Generally, the model captures the seasonal features of circulation in this region and has been used in previous studies (Cai, Liu, Liu, & Gan, 2022; Chu et al., 2022b; L. Cui, Liu, Chen, & Cai, 2024)."

[revised manuscript text omitted]

4. Introduction: mass accumulation dynamics in different sub-regions in the PRE is an important part of the current study. If available, please supplement more previous findings on the different material accumulation features in these regions (e.g., total organic carbon, pollutants, suspended sediments, or other water quality parameter), to emphasize why we should care about dividing the PRE into the six subregions. Statements in lines 145-146 could fit in this part. It also needs to mention and explain why this study used a model producing climatological hydrodynamics rather than focusing on years

with realistic forcings.

**Response:** Thanks for the suggestions.

In the previous studies of accumulations in the PRE, D. Zhang et al. (2013) found trace elements had the highest concentration in the western part of PRE. The microplastics concentrated in Hong Kong water and western part of the PRE (Lam et al., 2020). Tao, Niu, Dong, Fu, and Lou (2021) revealed that accumulation of nitrogen and silicate appeared in the upper part of PRE and phosphorus pollution was observed in the northeast of PRE (e.g., Shenzhen Bay). Suspended solids are prone to gather in the head of the PRE in winter while the west part of the PRE in summer (X. Zhang et al., 2013). The total organic carbon is more likely to concentrate in Macau, Shenzhen Bay and Hong Kong (Guo, Ye, & Lian, 2016).

The six subregions were defined based on the accumulation regions obtained from the tracking (Figure 3b, d in the original manuscript). Those regions have different accumulation pattern and well fit the mass/pollutant accumulation regions in the previous studies, such as UPPER and WEST PRE are in accord with the accumulation of suspended solids and trace elements in X. Zhang et al. (2013) and D. Zhang et al. (2013).

Before reproducing the results in a specific year, we want to figure out the basic response of the accumulation dynamics under typical forcing and circulation in the PRE, and how it varies due to the changes in the tidal current and river discharge. Thus, in this study, the climatological forcing and hydrodynamics that could represent the typical seasonal feature were used to eliminate the possible anomalies caused by synoptic or interannual signals.

Following reviewer's suggestions, we integrate the above information in the revised manuscript:

Line 61-71:

With intensified human activities, pollutant sinks related to the accumulation phenomena in the PRE have attracted attention. Tao et al. (2021) revealed that the upper part of the PRE is a target sink for nitrogen and silicate. D. Zhang et al. (2013) found that trace elements prefer to accumulate on the PRE's west side. Higher concentrations of microplastics have been observed in western estuaries and Hong Kong waters (Lam et al., 2020). Similarly, studies on hypoxia have shown that the convergence of buoyancy-driven currents and wind-driven shelf flows contributes to the formation of stable water columns, providing favorable conditions for the development of hypoxic zones (e.g., D. Li et al. (2021); X. Li et al. (2020)). The high frequency of hypoxia in the estuary during summer is related to the strong stratification of the water column (Y. Cui, Wu, Ren, & Xu, 2019; H. Zhang & Li, 2010). These accumulation patterns in the PRE are more concerned with the measurement of pollutant concentrations (Tao et al., 2021), estimation of the pollutant accumulation rate (L. Zhang et al., 2009), and discussion of the sources of pollutants (Ye, Huang, Zhang, Tian, & Zeng, 2012), and lack a discussion on the understanding of accumulation spatial patterns and underlying physical control.

5. Line 13: 'plume fronts' only occurred in the abstract. Please use a consistent description on fronts with the main text.

   **Response:** Thanks a lot. The 'plume fronts' will be changed into 'river discharges' in the revised abstract. The description on the front is unified in the revised manuscript.

6. Line 65: is the right panel of Figure 1 the model domain? If so, clearly state it in the caption. If not,

please add a figure showing the model domain.

**Response:** Thanks for reviewer's reminder. The details of model domain have clearly shown in Figure R1. The regions inside the red line in Figure R1 are the model domain. The yellow line is the seaside boundary to identify whether the particles has left the estuary.

7. Line 77: climate → climatology. Which period did the climatology forcing represent for?

**Response:** Thanks a lot. We will revise the typo. The climatology represents the mean forcing averaged between 1994 and 2018.

8. Lines 78-79: please supplement more detailed descriptions of the coarse resolution model or add related references.

**Response:** Thanks for reviewer's reminder. There is a well validated coarser model simulates the shelf current which covers entire NSCS. The results provide the barotropic and baroclinic velocities, temperature, salinity, sea level along the boundaries of PRE. This model was well validated and used in previous investigations (L. Cui, Cai, & Liu, 2023; L. Cui et al., 2024; Deng et al., 2022).

[Figure]

Figure R2. The domain of coarser model.

9. Line 79: clarify 'statistics of atmosphere forcing…'?

10. Line 80: what is the accuracy of the monthly river discharge rate by ECMWF in this region? Why not using the locally observed river discharge?

**Response:** Thanks for reviews' reminder. Those two comments are related to the model configuration, we have responded in point 1 above. And the details can be found in Line 99-104 in the revised manuscript.

Line 99-104: "Wind forcing, heat flux, and precipitation were obtained from ERA5 atmospheric reanalysis data from the European Center for Medium-Range Weather Forecasts (ECMWF) and were used to force ocean circulation through the implementation of the bulk computation algorithm (Fairall et al., 2003). The shelf current was obtained from a coarser model with good validation that can cover the North-South China Sea and provide information on the barotropic and baroclinic velocities, temperature, salinity, and sea level along the boundaries of the PRE (Deng et al., 2022)."

11. Line 87: what does 'other complicated hydrodynamic processes' imply? What is 'computations'? Computations of particle trajectory? Also, please add references to support the statement: 'The results of the computations…in the estuary-shelf system'.

**Response:** Thanks for the comment. The Lagrangian tracking model considers the advection, turbulence, individual behaviors of particles (e.g. vertical sinking, floating or swimming velocity), settlement and boundary behaviors during the particle trajectory simulations, which can support to obtain the approximate realistic trajectories in the real world with complicated hydrodynamic processes. The "computations" means the computation of the particle trajectories. This tracking module well captures the trajectories in previous studies (Chu et al., 2022a; Liang et al., 2021; North et al., 2011).

The sentences were refined to make it clear

Line 112-115: "To reasonably calculate the Lagrangian trajectories in the circulation of estuary-shelf systems, the tracking model considers the advection, turbulence, individual behaviors of particles (e.g., vertical sinking, floating, or swimming velocity), settlement, and boundary behaviors during particle trajectory simulations."

Line 110-112: "Particle trajectories were traced by a three-dimensional offline Lagrangian TRANSport model (LTRANS v.2b), which captures complicated dynamical processes in the real world using Eulerian flow fields and turbulent mixing from the hydrodynamic model (Chu et al., 2022a; Liang et al., 2021; North et al., 2011)."

12. Line 90: clarify 'different diffusivity coefficient' in the vertical and horizontal turbulence.

**Response:** Thanks for the comment that helps us to correct the misleading in this sentence. The diffusivity coefficient is provided by the hydrodynamical model. In the hydrodynamical simulation, the vertical diffusivity was calculated through Mellor-Yamada 2.5 turbulence-closure module coefficient (Mellor & Yamada, 1982), which is widely used in the coastal simulation (Choi, Park, Choi, Jung, & Kim, 2021; J. Liu, Lu, & Li, 2019; Robertson & Hartlipp, 2017). While in horizontal direction, the harmonic horizontal diffusion with uniform value was used. The diffusive coefficient in output data was used by LTRANS to introduce the random walk during simulation.

We revised the manuscript to make it clear.

Line 106-108: "Vertical turbulence and diffusion coefficient are determined by the Mellor-Yamada 2.5 turbulence-closure module (Mellor & Yamada, 1982), which provides the turbulent mixing coefficient."

13. Lines 91-93: the descriptions on particle tracking experiments are unclear. Were the 8386 particles released each day in January and July? How was the 'uniform release' achieved, i.e., what are the vertical and horizontal interval of each particle? How will the particle behave when they reach the boundaries, including open, land, surface, and bottom boundaries? time step of particle tracking? output frequency of the history files that were used to drive the particle tracking and the output frequency of particle location?

**Response:** Thanks for the comments.

For the surface tracking case, 8386 particles were released at the water surface, which are uniformly distributed covering the entire estuary and the adjacent shelf with a 0.01degree horizontal interval.

The particles were released every two days and tracked for 30 days after being released. The bottom tracking case is same as surface one but releasing at the bottom.

To drive the particle tracking in summer and winter, results from hydrodynamic simulation were saved each 20 minutes in January and July, respectively. During the tracking of trajectories, the time step of particle tracking is 30 seconds and the output frequency of the particles' location is 20 minutes. When particles reach the land, surface and bottom boundaries, it would be reflected back to the last step location and wait a suitable current to take it away. If they reach the boundaries of the model domain, the particles would only remain the last locations' information.

In the revised manuscript, we clarify the descriptions on the particle tracking experiments.

Line 122-126: "In the surface/bottom tracking case, 8386 particles were uniformly distributed at water surface/bottom across the estuary and adjacent shelf with a 0.01degree interval. Particles were released every two days and tracked for 30 days. The hydrodynamic simulation results were stored every 20 min in January and July to drive the particle tracking during summer and winter, respectively. During trajectory tracking, the time step of particle tracking was 30 seconds, and their locations were recorded every 20 minutes."

14. Line 105: it should be nit0

**Response:** Corrected and thanks

15. Line 106: it is pt in equation (1). Please use the same symbol.

**Response:** Thanks for reviewers' suggestion. The expression of the formula was carefully checked in the revised manuscript, and we will standardize the symbol in the formula.

16. Line 121: 'transport accumulation' seems to be an odd expression.

**Response:** Thanks a lot. Actually, we want to express accumulation pattern here, perhaps our expression is not accurate. We will change it into 'Accumulation pattern and regional connectivity' in Line 150.

17. Lines 123-124: did particles come back after moving out of the seaside boundary?

**Response:** When the particles move out of the seaside boundary of the estuary (the yellow line in Figure R1), it has the probability to return to the study area under the effect of tidal currents and wind forcing. If the particles reach the model domain, they will not come back again. We clarify this in the revised manuscript:

Line 155-157: "If particles move beyond the seaside boundary of the estuary, they may return to the study area due to tidal currents and wind forcing. However, once the particles reached the model domain, they will not be backed again."

18. Line 135: particles were only tracked for 30 days as stated in section 2.1. Why would the maximum day in x-axis 60 days?

**Response:** Thanks for reviewers' reminder. To choose a suitable time to explore the particle accumulation, we designed a test to get the trajectories for 60 days. As shown in Figure 2 in the manuscript, after 30 days, the most particles leave the estuary and the percentage decreased to less than 10%. Thus, in the following analysis, only the 30 days results were used to check the accumulation dynamics.

19. Line 148: What leads to the further landward intrusion of bottom water in winter?

**Response:** we select a typical transect in the PRE (Transect AB in Figure R4b) to show the changes of vertical structure of circulation. In winter, an obviously strengthened landward current is observed at the bottom, which is favorable for the bottom intrusion (Figure R3). The river discharge is larger in summer than that in winter in the PRE (Harrison, Yin, Lee, Gan, & Liu, 2008), and a strong outflow pushes the water to the lower estuary and hinders the intrusion of shelf water into the estuary. In winter, less river discharge leads to a weaker resistance to the intrusion water, which causes strengthened landward movement of the bottom water and pushes the isopycnals to move into the inner estuary. Thus, although the front is similar (Figure 6 in the original manuscript), the gravitational circulation induced by river discharge contributes to landward movement of water accumulation in winter.

[Figure]

Figure R3. The flow field of standard in transect AB at the bottom layer in summer (a) and winter (b), respectively. The colorbar represents the onshore velocity, the positive value means the landward motion in the transect AB. The arrow represents the current direction and the length is the strength of currents. The red line represents the isopycnal of 1010 kg/m⁻³ and 1015 kg/m⁻³.

20. Line 155. Does probability have a unit? percentage? It is not easy to see the difference in the gray color for velocity arrows. Suggest using arrow length to represent the magnitude of velocity. In addition, there are some unnecessary gray dividing lines in Figure 3 and figures below. Please remove them.

**Response:** Thanks for reviewers' reminder. The accumulation probability is calculated based on the percentage and does not have a unit.

The flow field will be separated from Figure 3 in the original manuscript and show below, and a more obvious colormap will be used to represent the velocity changes in the new version of flow field. The gray dividing lines in Figure 3 in the original manuscript will also be removed.

[Figure]

Figure R4. The flow field at the surface layer (a) and bottom layer (b) during summer time. (c-d) are same as (a-b), except for during winter time. The colorbar represents the velocity of current.

21. Line 182: the accumulation probability was averaged over each sub-domain and not differentiate between surface and bottom in Figure 5? The div(V) in Figure 5c-d shows surface or bottom or depth-averaged results?

**Response:** Thanks for reviewers' comments. The accumulation probability in Figure 5 in original manuscript is averaged in bottom of each subregion. As shown in Figure R5, the bottom has a much larger accumulation, while the surface features with the strong offshore transport, as Lam et al. (2020) founded higher concentration of microplastics at the bottom layer than at the surface layer in the PRE. Therefore, we chose the bottom layer to discuss the internal connection and accumulation dynamics between different regions in the PRE. The accumulation and DIV were calculated at the bottom layer. We clarify this in the manuscript:

Line 227-230: "The bottom divergence of the horizontal current, i.e., $\nabla_h \vec{V}_h = \frac{\partial u}{\partial x} + \frac{\partial v}{\partial y}$, which $u$ and $v$ represent the bottom zonal and meridional velocity, is calculated to examine its influence on the identified bottom accumulation regions (Fig. 7a, b). We established a connection between the average accumulation probability in each subregion and the divergence of the horizontal current."

Line 239-240: "Spatial distribution of $\nabla_h \vec{V}_h$ also illustrates that the intensified negative values occur in the region of the high accumulation probability (Fig. 7c-d)"

[Figure]

Figure R5: (a–b) Accumulation probability (color, $D^{t_0}$ in Eq. (2)) at the surface layer and bottom layer during summer time, respectively. The color bar indicates the magnitude of the accumulation probability. (c-d) is the same as (a-b) but winter time.

22. Line 215: the unit of density front should be kg m-4 from the equation in line 205.

**Response:** Thanks for reviewers' kindly reminder, we corrected the typo in the revised manuscript.

23. Section 3.3: please provide more details on the testing cases (or put them in the supplementary). For example, which options or files in roms were removed for the case removing tidal currents. How many rivers were included in the model domain? A time series of river discharge in base case and test case will be useful information.

**Response:** Thanks for reviewers' suggestions.

In the ROMs, the wind forcing, river discharge and tidal current were provided by different forcing files. The model read those forcing files during the simulation. In the sensitivity test, the configuration was the same but the magnitude of the wind speed, river discharge rate and tidal current were changed respectively. In the sensitivity case with reduced river discharges, the magnitude of river discharge rate is decreased to 20% of the control run. In the no tide case, the tide induced current and elevation is not included along the open boundary conditions.

To make the configuration of the model clear, we added the supplementary file to introduce the monthly river discharges and seasonal monsoon (Figure R6). The configuration of the sensitivity test was also clarified in the revised manuscript.

Line 269-271: "In the case of reduced river discharge, the magnitude of river discharge in the forcing file was reduced by 20%. The magnitude of the tidal current was set to zero to remove the tidal currents."

[Figure]

Figure R6. (a-b) The seasonal mean wind speed (m/s) during summer (June–August) and winter (December–February), respectively. (c) The monthly river discharge in the PRE.

24. Line 244: if the particle tracking was conducted for a longer time, will MO accumulate water from other regions?

**Response:** Thanks a lot. Since the particles will finally leave the estuary (Figure 2 in the original manuscript), the longer tracking time does not make MO accumulate water from other regions. We have a test with a longer tracking period of 30 days, the basic accumulation pattern and the connection among the different regions are similar.

25. Line 245: the arrows in Figure 7c-d seems to be confusing. Not every regions were noted by arrows? What does different arrow directions represent?

**Response:** Thanks for reviewers' reminder. We only choose the region with a larger accumulation anomaly to plot the arrows in the figure. The direction of the figure represents the changes of the accumulation and divergence in each sensitivity test. Generally, in those cases we noticed a clear negative relationship between the accumulation and divergence. For example, in the no tide case during summer time (Figure R7), the left-upward arrow of the EAST point represents the strengthened of net convergence (leftward) and increased accumulation (upward).

In the revised manuscript we make it clear that the arrows are plotted mainly in the regions with significant accumulation changes.

Line 302-304: "The arrow represents the changes of $\nabla_h \vec{V}_h$ and accumulation due to the removal of tidal currents in the region, with significant changes in the accumulation."

[Figure]

Figure R7: (a-b) Scatter plot of accumulation probability against $\nabla_h \vec{V}_h$ for various subregions during summer and winter in removing tide cases. The arrow represents the changes of $\nabla_h \vec{V}_h$ and accumulation due to the removing of tide in each subregion. (c-d) The connection between six regions for removal of tidal current at the bottom layer during summer and winter time, respectively.

26. Line 281: suggest adding some descriptions on wind strength and magnitude of river discharge in the main text.

**Response:** Thanks a lot. According to reviewers' comments, some basic information of wind forcing and river discharge in the PRE are supplemented in Line26-47 in the revised manuscript, which can provide a better background of the PRE.

Line 26-47:

[revised manuscript text omitted]

Pan, J., Lai, W., & Thomas Devlin, A. (2020). Circulations in the Pearl River Estuary: Observation and Modeling. In *Estuaries and Coastal Zones - Dynamics and Response to Environmental Changes*.

Robertson, R., & Hartlipp, P. (2017). Surface wind mixing in the Regional Ocean Modeling System (ROMS). *Geoscience Letters, 4*(1), 24. doi:10.1186/s40562-017-0090-7

Shchepetkin, A. F., & McWilliams, J. C. (2005). The regional oceanic modeling system (ROMS): a split-explicit, free-surface, topography-following-coordinate oceanic model. *Ocean modelling (Oxford), 9*(4), 347-404. doi:10.1016/j.ocemod.2004.08.002

Song, Y., & Haidvogel, D. (1994). A Semi-implicit Ocean Circulation Model Using a Generalized Topography-Following Coordinate System. *Journal of Computational Physics, 115*(1), 228-244. doi:https://doi.org/10.1006/jcph.1994.1189

Tao, W., Niu, L., Dong, Y., Fu, T., & Lou, Q. (2021). Nutrient Pollution and Its Dynamic Source-Sink Pattern in the Pearl River Estuary (South China). *Frontiers in Marine Science, 8*. doi:10.3389/fmars.2021.713907

Wong, L. A., Chen, J. C., Xue, H., Dong, L. X., Su, J. L., & Heinke, G. (2003). A model study of the circulation in the Pearl River Estuary (PRE) and its adjacent coastal waters: 1. Simulations and comparison with observations. *Journal of Geophysical Research: Oceans, 108*(C5). doi:https://doi.org/10.1029/2002JC001451

Ye, F., Huang, X., Zhang, D., Tian, L., & Zeng, Y. (2012). Distribution of heavy metals in sediments of the Pearl River Estuary, Southern China: Implications for sources and historical changes. *Journal of Environmental Sciences, 24*(4), 579-588. doi:https://doi.org/10.1016/S1001-0742(11)60783-3

Zhang, D., Zhang, X., Tian, L., Ye, F., Huang, X., Zeng, Y., & Fan, M. (2013). Seasonal and spatial dynamics of trace elements in water and sediment from Pearl River Estuary, South China. *Environmental Earth Sciences, 68*(4), 1053-1063. doi:10.1007/s12665-012-1807-8

Zhang, H., & Li, S. (2010). Effects of physical and biochemical processes on the dissolved oxygen budget for the Pearl River Estuary during summer. *Journal of marine systems, 79*(1), 65-88. doi:https://doi.org/10.1016/j.jmarsys.2009.07.002

Zhang, L., Yin, K., Wang, L., Chen, F., Zhang, D., & Yang, Y. (2009). The sources and accumulation rate of sedimentary organic matter in the Pearl River Estuary and adjacent coastal area, Southern China. *Estuarine, coastal and shelf science, 85*(2), 190-196. doi:https://doi.org/10.1016/j.ecss.2009.07.035

Zhang, X., Shi, Z., Liu, Q., Ye, F., Tian, L., & Huang, X. (2013). Spatial and temporal variations of picoplankton in three contrasting periods in the Pearl River Estuary, South China. *Continental shelf research, 56*, 1-12. doi:https://doi.org/10.1016/j.csr.2013.01.015

Zu, T., & Gan, J. (2015). A numerical study of coupled estuary–shelf circulation around the Pearl River Estuary during summer: Responses to variable winds, tides and river discharge. *Deep Sea Research Part II: Topical Studies in Oceanography, 117*, 53-64. doi:https://doi.org/10.1016/j.dsr2.2013.12.010

Zu, T., Gan, J., & Erofeeva, S. Y. (2008). Numerical study of the tide and tidal dynamics in the South China Sea. *Deep Sea Research Part I: Oceanographic Research Papers, 55*(2), 137-154. doi:https://doi.org/10.1016/j.dsr.2007.10.007

Zu, T., Wang, D., Gan, J., & Guan, W. (2014). On the role of wind and tide in generating variability of Pearl River plume during summer in a coupled wide estuary and shelf system. *Journal of marine systems, 136*, 65-79. doi:https://doi.org/10.1016/j.jmarsys.2014.03.005

---

## Referee Report (RR1)

I appreciate the efforts the authors have made to reply to my comments. I have several suggestions for the clear version of the manuscript.

(1) the correct format of the horizontal velocity divergence should be $\nabla_h \cdot \vec{V_h}$ (?)

(2) Section 2.1 can be reorganized to describe model (and its validation) first and then the details of particle tracking.

(3) Line 204, is it 'after 30 d' rather than 'after 20 d'?

(4) correct the label in panel (d) of Figure 5 and the labels in the lower panels of Figure 7.

(5) Line 250: PRE was introduced as a partially mixed estuary in Line 35 but here it states "as a salt-wedge estuary".

(6) correct 'river discharges' in the caption of Figure 10 to be 'tide'.

(7) Figure 15: what are these dashed arrows?

(8) In the supplementary, please add the full name of 'MUR'. Figure S3 is not mentioned in the main text.

(9) The authors may consider put some figures into the supplementary, such as those velocity field.

(10) More efforts could be made to improve the languages.

---

## Referee Report (RR2)

**Second review for the manuscript**

**'Exploring water accumulation dynamics in the Pearl River Estuary from a Langrangian Perspective'.**

The authors have significantly improved the quality of their manuscript and have responded satisfactorily in most comments and provided appropriate clarifications wherever needed. However, there are still some ambiguities which are important to be addressed before the paper is published. In addition, I provide some comments that hopefully the authors would find useful for improving their layout.

**Major comment:**

There is a concern regarding the use of the term probability in the figures which requires clarification. In figure 5, the authors plot $D$ (and not $D^{t_0}$ ) which is the evolution of the initial distribution but not probability. In any case, the probability cannot be above 1. I would advise to either replace the word probability or normalize the results (e.g., by dividing with the total number of particles) so that the values remain below 1.

Do the authors plot in Figure 6 accumulation probability? And is it the same with what is plotted in Figure 7? Because there seems to be a disagreement between what is plotted in figure 6 and figure 7. In figure 6, the authors plot the probability of particles moving in each region and the range of values is between 0 and 0.07. But then, in Figure 7 a and b, the range of values of accumulation probability extends between 0 and 4. How are these two figures related? Do they show both the same thing (i.e., accumulation probability)? The same concern about the probability being more than 1 applies here.

Also, in Figure 6, the authors mention in the caption that the plot shows the connection between six subregions, but the legend shows probability. Please clarify these terms and modify the caption accordingly.

Similarly, in Figure 10 and 13 the authors mention in the captions that they plot probability anomaly. I find the word probability again irrelevant, at least based on their definition of anomaly as given in their response. Besides, probability cannot be negative. It would probably be better to remove the word probability from these figures.

**Minor:**

1. To reduce the number of figures in your paper, I suggest the following:
- Merge the panels of Figure 3 and Figure 5
- Figure 4 can be moved in the supplementary
- Merge the panels of Figure 10 and Figure 13
2. Please add a sentence in your manuscript to describe how you define the anomaly in Figure 10 and Figure 13.
3. Please add a sentence in your manuscript with the explanation you give on your response on why you decided to focus on the bottom layers accumulation only.

Furthermore:

Line 48 state instead of 'health'

Line 52 biogeochemical conditions instead of 'health'

Line 57 usually appear eutrophic.

Line 60 add space between sinks and (Mestres)

Line 62 remove D from Zhang

Line 68-71 this sentence is not very well written, please rephrase.

Line 97 layers instead of levels.

Line 101-102 I would advise to include Figure R8 in the Supplementary

Line 112 remove Elizabeth NEW

General comment: do not include authors' first name when citing papers in the manuscript.

---

## Author Response (AR2)

**Response 1**

**I appreciate the efforts the authors have made to reply to my comments. I have several suggestions for the clear version of the manuscript.**

Thanks for review's kindly suggestion, we will revise and response for follow comments point by point:

(1) the correct format of the horizontal velocity divergence should be $\nabla_h \overrightarrow{V_h}$ (?)

Response: The correct format of the horizontal velocity divergence is $\nabla_h \cdot \overrightarrow{V_h}$, which is

calculated by $\nabla_h \cdot \overrightarrow{V_h} = (\frac{\partial}{\partial x}\vec{\imath} + \frac{\partial}{\partial y}\vec{\jmath})(u\vec{\imath} + v\vec{\jmath}) = \frac{\partial u}{\partial x} + \frac{\partial v}{\partial y}$.

(2) Section 2.1 can be reorganized to describe model (and its validation) first and then the details of particle tracking.

**Response:** Thanks for reviewer's suggestion. Section 2.1 was reorganized to describe the model (with model validation) first then the details of particle tracking.

**Line 108-111:** "This model, based primarily on climatological data, was carefully verified using satellite remote sensing and long-term observations to ensure an accurate representation of the hydrodynamic properties (Fig. S3, 4). Overall, the model accurately captured the seasonal variability of the hydrodynamic features in this region and has been used in previous studies (Cai, Liu, Liu, & Gan, 2022; Chu et al., 2022b; Cui, Liu, Chen, & Cai, 2024)."

(3) Line 204, is it 'after 30 d' rather than 'after 20 d'?

**Response:** Thanks for reviewer's reminder. We mentioned that almost 80% of the particles will leave the estuary seaside boundary with 20-day. Thus, we adopt the transition matrix of 20-day results in this paper to show the details of particle mass in estuary. The typo in '30-day' have corrected into '20-day' in the revised manuscript.

**Line 122-123:** "Particles were released every two days and tracked for 20 days."

**Line 168-170:** "Using the trajectories of the released particles within 20 days, we explored the final evolved state, which is used to quantify the accumulation targets, as a result of the complex hydrodynamics of estuarine circulation."

(4) correct the label in panel (d) of Figure 5 and the labels in the lower panels of Figure 7.

**Response:** Thanks very much. We have corrected the typo in the revised paper.

[Figure]

Figure R2: (a–b) Particle mass (color, $D^t$ in Eq. (2)) at the surface layer and bottom layer during summer time, respectively. The color bar indicates the magnitude of the particle mass, higher value represents stronger accumulation. (c-d) is the same as (a-b) but winter time.

(5) Line 250: PRE was introduced as a partially mixed estuary in Line 35 but here it states "as a salt-wedge estuary".

**Response:** Thanks for the comments. we corrected it as partially mixed estuary.

**Line 34-35:** "The PRE is a partially mixed estuary in which circulation is jointly controlled by river discharge, tides, wind, and topography."

**Line 247-249:** "The existence of a salinity front acts as a barrier to particle transport and plays an important role in accumulation regions, such as coarse particles will accumulate at the bottom salinity front (Defontaine et al., 2020; He et al., 2018; Vermeiren et al., 2016)."

(6) correct 'river discharges' in the caption of Figure 10 to be 'tide'.

**Response:** Thanks for reviewer's reminder. We have corrected the typo in the revised manuscript.

**Line 290-291:** "Figure 8: (a-b) Particle mass ($D^t$) anomaly in the removing tide current case during summer and winter, respectively. A negative value represents the strengthened offshore transport without tidal current."

(7) Figure 15: what are these dashed arrows?

**Response:** Thanks a lot. The dashed arrows in Figure 15 in previous manuscript are same with the solid arrows, which represent the water transport directions in different

season in the PRE. We corrected them both into solid arrows in the revised manuscript.

[Figure]

Figure R3: The accumulation connections schematic in the PRE during summer (red arrow) and winter time (green arrow). The map color in red represents the high accumulations in summer, while green represented winter. The star indicates that the tide dominated the current, and the triangle represents river discharge.

(8) In the supplementary, please add the full name of 'MUR'. Figure S3 is not mentioned in the main text.

**Response:** Thanks for the comment. The full name of 'MUR' is 'Multiple-scale Ultra-high Resolution', we have added in the supplementary. And Figure S3 in previous supplementary is mentioned in Line 119-120 in the revised manuscript.

**Line 9-11 in the supplementary:** "Figure S3: (a-b) Climatological Sea Surface Temperature (SST) anomaly during summer and winter from the Multiple-scale Ultra-high Resolution (MUR) SST reanalysis product from the Jet Propulsion Laboratory (JPL) of NASA (2002-2021). (c-d) are the same as (a-b) but for the model results."

(9) The authors may consider put some figures into the supplementary, such as those velocity field.

**Response:** Thanks for reviewer's suggestions. We have put all velocity field in the supplementary in the revised version, the related description in the manuscript has revised.

(10) More efforts could be made to improve the languages

**Response:** Thanks for reviewer's comments. We have carefully checked the languages in the manuscripts and improved some expression.

**Reference:**

Cai, Z., Liu, G., Liu, Z., & Gan, J. (2022). Spatiotemporal variability of water exchanges in the Pearl River Estuary by interactive multiscale currents. *Estuarine, Coastal and Shelf Science, 265*, 107730. doi:https://doi.org/10.1016/j.ecss.2021.107730

Chu, N., Liu, G., Xu, J., Yao, P., Du, Y., Liu, Z., & Cai, Z. (2022). Hydrodynamical transport structure and lagrangian connectivity of circulations in the Pearl River Estuary. *Frontiers in Marine Science, 9*. doi:10.3389/fmars.2022.996551

Cui, L., Liu, Z., Chen, Y., & Cai, Z. (2024). Three-Dimensional Water Exchanges in the Shelf Circulation System of the Northern South China Sea Under Climatic Modulation From ENSO. *Journal of Geophysical Research: Oceans, 129*(4), e2023JC020290. doi:https://doi.org/10.1029/2023JC020290

Defontaine, S., Sous, D., Tesan, J., Monperrus, M., Lenoble, V., & Lanceleur, L. (2020). Microplastics in a salt-wedge estuary: Vertical structure and tidal dynamics. *Marine pollution bulletin, 160*, 111688. doi:https://doi.org/10.1016/j.marpolbul.2020.111688

He, Q., Zhan, H., Cai, S., He, Y., Huang, G., & Zhan, W. (2018). A New Assessment of Mesoscale Eddies in the South China Sea: Surface Features, Three-Dimensional Structures, and Thermohaline Transports. *123*(7), 4906-4929. doi:https://doi.org/10.1029/2018JC014054

Lu, Z., & Gan, J. (2015). Controls of seasonal variability of phytoplankton blooms in the Pearl River Estuary. *Deep Sea Research Part II: Topical Studies in Oceanography, 117*, 86-96. doi:https://doi.org/10.1016/j.dsr2.2013.12.011

Vermeiren, P., Muñoz, C. C., & Ikejima, K. (2016). Sources and sinks of plastic debris in estuaries: A conceptual model integrating biological, physical and chemical distribution mechanisms. *Marine pollution bulletin, 113*(1-2), 7-16. doi:10.1016/j.marpolbul.2016.10.002

Wong, L. A., Chen, J. C., Xue, H., Dong, L. X., Su, J. L., & Heinke, G. (2003). A model study of the circulation in the Pearl River Estuary (PRE) and its adjacent coastal waters: 1. Simulations and comparison with observations. *Journal of Geophysical Research: Oceans, 108*(C5). doi:https://doi.org/10.1029/2002JC001451

<h1 style="text-align:center;color:red">Response 2</h1>

**Second review for the manuscript**
**'Exploring water accumulation dynamics in the Pearl River Estuary from a Lagrangian Perspective'.**
The authors have significantly improved the quality of their manuscript and have responded satisfactorily in most comments and provided appropriate clarifications wherever needed. However, there are still some ambiguities which are important to be addressed before the paper is published. In addition, I provide some comments that hopefully the authors would find useful for improving their layout.

**Response:** Thanks for reviewer's perceptive suggestions, we carefully revised our manuscript based on your comments and listed our responses as follow.

**Major comment:**
1. There is a concern regarding the use of the term probability in the figures which requires clarification. In figure 5, the authors plot $D$ (and not $D^{t_0}$) which is the evolution of the initial distribution but not probability. In any case, the probability cannot be above 1. I would advise to either replace the word probability or normalize the results (e.g., by dividing with the total number of particles) so that the values remain below 1.

   **Response:** Thanks for the comment which helps us to further refine our manuscript. Since this and the following concerns are mainly related the calculation during Markov Chains and the meaning of different figures, we plot a schematic figure (R1) to clarity it.

[Figure]

Figure R1: The schematic in the calculation of the Markov Chains.

As shown in Figure R1, the $D^{t_0}$ represents the initial mass distribution, and its evolution was calculated from the multiplication of $D^{t_0}$ with the transition matrix $P^t$, i.e. $D^t = D^{t_0} \times P^t$. Here the $P^t$ is derived from the multiplication of the

probability matrix $p^t$ within each time interval that $[t_0, t_0 + \Delta t]$,$[ t_0 + \Delta t, t_0 + 2\Delta t]$, … $[t_0 + T - \Delta t, t_0 + T]$. The $D^t$ is the mass distribution in future state and $P^t$ represents net probability trend between different regions in the study area, area with high value in $D^t$ indicates the strong accumulation target. Using the final $D^t$, we obtained the correlation between the regional particle mass $D^t$ and the horizontal current divergence $\nabla_h \cdot \overrightarrow{V_h}$,

In the Figure 5 in previous manuscript, as mentioned by reviewer, the $D^t$ was used, thus "probability" is not a proper description. Since some regions attract more particles from other regions, the value could be above 1. Following reviewer's reminder, we corrected the caption and clarified the calculation of the Markov Chains.

**Line 138-142:** "The $D^t$ is the evolution of the initial condition under complicated hydrodynamic motion, which calculated from the multiplication of the $D^{t_0}$ with the transition matrix $P^t$ (Fig. 2). The transition matrix $P^t$ is derived from the multiplication of the probability matrix $p^t$, illustrate the net probability trend between different regions in the study area. Areas with high values in $D^t$ act as strong accumulation targets of particles."

**Line 197-198:** "Figure 4: Particle mass (color, $D^t$) at the surface layer and bottom layer during summer time, respectively. Higher value represents stronger accumulation. (c-d) is the same as (a-b) but during winter time."

2. Do the authors plot in Figure 6 accumulation probability? And is it the same with what is plotted in Figure 7? Because there seems to be a disagreement between what is plotted in figure 6 and figure 7. In figure 6, the authors plot the probability of particles moving in each region and the range of values is between 0 and 0.07. But then, in Figure 7 a and b, the range of values of accumulation probability extends between 0 and 4. How are these two figures related? Do they show both the same thing (i.e., accumulation probability)? The same concern about the probability being more than 1 applies here. Also, in Figure 6, the authors mention in the caption that the plot shows the connection between six subregions, but the legend shows probability. Please clarify these terms and modify the caption accordingly.

**Response:** Thanks for the comments and apology for the misleading in the captions.
In Figure 6, the $P^t$ (Figure R1) was plotted, which shows the probability of particles moving in each region. We used it to represent the connectivity between different regions. While in Figure 7, the $D^t$ was used illustrate the negative correlation with the net $\nabla_h \cdot \overrightarrow{V_h}$, thus the y axis ranges between 0 and 4. As reviewer reminded, they are not the same thing. In the revised manuscript, the description

of the calculation of Figure 6 and Figure 7 in previous manuscript are added, and the caption of Figure 7 in previous manuscript are revised.

**Line 183:** "Figure 4 shows the $D^t$, in which regions of high value represent the favorable targets for particles accumulation."

**Line 206-207:** "Subsequently, using the trajectories, the transition matrix ($P^t$) among each region during the tracking period is examined (Fig. 5)."

**Line 227-228:** "We established a connection between the average $D^t$ in each subregion and the divergence of the horizontal current $\nabla_h \cdot \vec{V_h}$."

**Line 244-245:** "Figure 6: (a-b) Scatter plot of regional $D^t$ against $\nabla_h \cdot \vec{V_h}$ for various subregions during summer and winter, respectively.".

3. Similarly, in Figure 10 and 13 the authors mention in the captions that they plot probability anomaly. I find the word probability again irrelevant, at least based on their definition of anomaly as given in their response. Besides, probability cannot be negative. It would probably be better to remove the word probability from these figures.
**Response:** Thanks for your advice. Figures 10 and 13 in previous manuscript are the anomaly of $D^t$, not the probability. We have corrected the caption as:

**Line 290:** "Figure 8: (a-b) Particle mass ($D^t$) anomaly in the removing tide current case during summer and winter, respectively."

**Line 329:** "Figure 10: (a-b) Particle mass ($D^t$) anomaly in the reducing river discharge case during summer and winter, respectively."

**Minor:**
1. To reduce the number of figures in your paper, I suggest the following:
- Merge the panels of Figure 3 and Figure 5
- Figure 4 can be moved in the supplementary
- Merge the panels of Figure 10 and Figure 13

**Response:** Thanks for your suggestions. Considering the two reviewer's suggestions, we modified figures in the previous manuscript as follow:
-Moved the figures of velocity field into the supplementary: Figure 3, Figure 9a-b, and Figure 12a-b.
-Merge the figure of vertical structure with the figure of particle mass: Figure 4 with Figure 5, Figure 9c-d with Figure 10, and Figure 12c-d with Figure 13.

2. Please add a sentence in your manuscript to describe how you define the anomaly in Figure 10 and Figure 13.

**Response:** Thanks for the suggestions. We added the description of anomaly in the manuscript.

**Line 280:** "Anomaly of $D^t$ between the case without tides and standard case, i.e., $D^t_{no-tide} - D^t_{standard}$ is compared in Figure 8."

**Line 319-320:** "Similar with Figure 8, the anomaly results are calculated by using the $D^t$ of case with reduced river discharges to subtract the $D^t$ of the standard case (Fig. 10a, b)."

3. Please add a sentence in your manuscript with the explanation you give on your response on why you decided to focus on the bottom layers accumulation only.

**Response:** Thanks for the comments. We added the sentence in Line 202-204 in the revise manuscript.

**Line 202-204:** "Compared to the bottom layer, the quicker motion at the surface layer cannot distinctly reveal an accumulation pattern. Hence, the accumulation pattern and regional connectivity are focused on the bottom layer."

**Furthermore:**
1. Line 48 state instead of 'health'
**Response:** corrected.
2. Line 52 biogeochemical conditions instead of 'health'
**Response:** corrected.
3. Line 57 usually appear eutrophic.
**Response:** corrected.
4. Line 60 add space between sinks and (Mestres)
**Response:** added and thanks.
5. Line 62 remove D from Zhang
**Response:** corrected.
6. Line 68-71 this sentence is not very well written, please rephrase.
**Response:** corrected and thanks.
7. Line 97 layers instead of levels.
**Response:** corrected.
8. Line 101-102 I would advise to include Figure R8 in the Supplementary
**Response:** added and thanks.
9. Line 112 remove Elizabeth NEW
**Response:** corrected.

**General comment:** do not include authors' first name when citing papers in the

manuscript.

**Response:** Thanks for reviewer's suggestion. We have checked the citation and corrected them in the revised manuscript.